# Programmable multimodal actuation in cholesteric liquid crystal elastomer hollow fibers beyond mechanochromism

Jiazhe Ma[1], John S. Biggins [2] ✉, Fan Feng [3] ✉ & Zhongqiang Yang [1,4] ✉

Cholesteric liquid crystal elastomers (CLCEs) change color under strain, offering attractive prospects for soft robotics and photonic devices. However, the helical structure of CLCEs averages out the exceptional anisotropy and soft elasticity of the nematic phase, leaving little scope for also using the director orientation to program their thermal or mechanical actuation. Here, we develop programmable CLCE hollow fibers with longitudinal, circumferential, or twisted alignments via the integration of dynamic boronic ester bonds and mechanical force/pressure-induced orientation, all while preserving sufficient periodicity for structural color. Upon inflation, these fibers exhibit diverse motions—expansion, contraction, elongation, twisting—with synchronous color adaptation. Accordingly, we derive a membrane balloon model based on the non-ideal neo-classical LCE energy with suitable CLCE director profiles, successfully capturing key mechanical features including non-monotonicity and sub-criticality. This study provides a paradigm for the development of intelligent shape- and color-changing systems in a bespoke and versatile way.

Liquid crystal elastomers (LCEs) are one of the most promising classes of smart soft materials, with a range of dramatic and useful properties arising from their combination of the mobile and switchable anisotropy of liquid crystals (LCs) and the large deformation mechanics of elastomers[1–4]. Most famous are nematic LCEs, which exhibit muscular contractile actuation along their alignment axis (director) upon heating through the nematic/isotropic phase transition, and also exotic "soft elasticity" in which some deformations can be accommodated by director rotation within the solid at almost zero elastic stress/energy[5,6]. Conversely, chiral nematic (cholesteric) elastomers (CLCEs) are known for their vivid structural colors and mechanochromism[7–9]. More precisely, the director in CLCEs follows the same periodic helical nanostructure as found in liquid cholesterics, leading to strong Bragg reflection of light with a wavelength $\lambda$ determined by the pitch. In CLCEs, these helices are embedded in an elastic solid, which additionally enables pitch, and hence color, to be manipulated mechanically[10–15]. CLCEs are thus very promising for applications in various fields such as sensors[16–19], information encryption[20,21], and biomimetic camouflage[22–24].

However, since a cholesteric helix samples all planar orientation directions equally, the price paid for these dramatic colors is that mechanically, CLCEs are overall isotropic in plane. CLCEs thus lack the actuation and global soft elasticity of simple nematic LCEs, and also cannot benefit from "director design" in which director patterns are used to sculpt complex shapes[25] or complex soft-elastic mechanical responses[26]. CLCEs are thus something of a "one-trick pony", offering color changes under mechanical strain, but little opportunity to control or produce the deformations themselves.

Pneumatic actuation, with its easy implementation and dramatic mechanical/shape change performance, has rapidly established itself as a facile and effective platform for soft robotics[27,28]. Inflatable nematic balloons/tubes/fibers, for instance, have been shown theoretically and experimentally to exhibit versatile actuation motions depending on their initial alignment, and also exceptional work

[1]Key Lab of Organic Optoelectronics and Molecular Engineering of Ministry of Education, Department of Chemistry, Tsinghua University, Beijing, People's Republic of China. [2]Department of Engineering, University of Cambridge, Cambridge, UK. [3]School of Mechanics and Engineering Science, Peking University, Beijing, People's Republic of China. [4]Laboratory of Flexible Electronics Technology, Tsinghua University, Beijing, People's Republic of China. ✉e-mail: jsb56@cam.ac.uk; fanfeng@pku.edu.cn; zyang@tsinghua.edu.cn

capacities[29–35]. In contrast, recent studies on inflatable CLCEs, while they do exhibit attractive blueshifts, remain confined to color changes without functional deformation, which sacrifices the versatility and designability of actuation motions[22,36,37]. Therefore, precisely programming the alignments of CLCEs is essential for exploring the tuning of both deformation pathways and colors guided by combining soft elasticity and pneumatic actuation.

Overcoming such a challenge promises unprecedented multimodal responsiveness in CLCE systems.

Drawing these ideas together, here we report an anisotropic deswelling-assisted template method to fabricate CLCE fibers with hollow geometries and structural colors (Figs. 1a, 1b, and 1c), then use thermo-activated exchange of boronic ester bonds, along with mechanical or pneumatic forces, to program such fibers with global

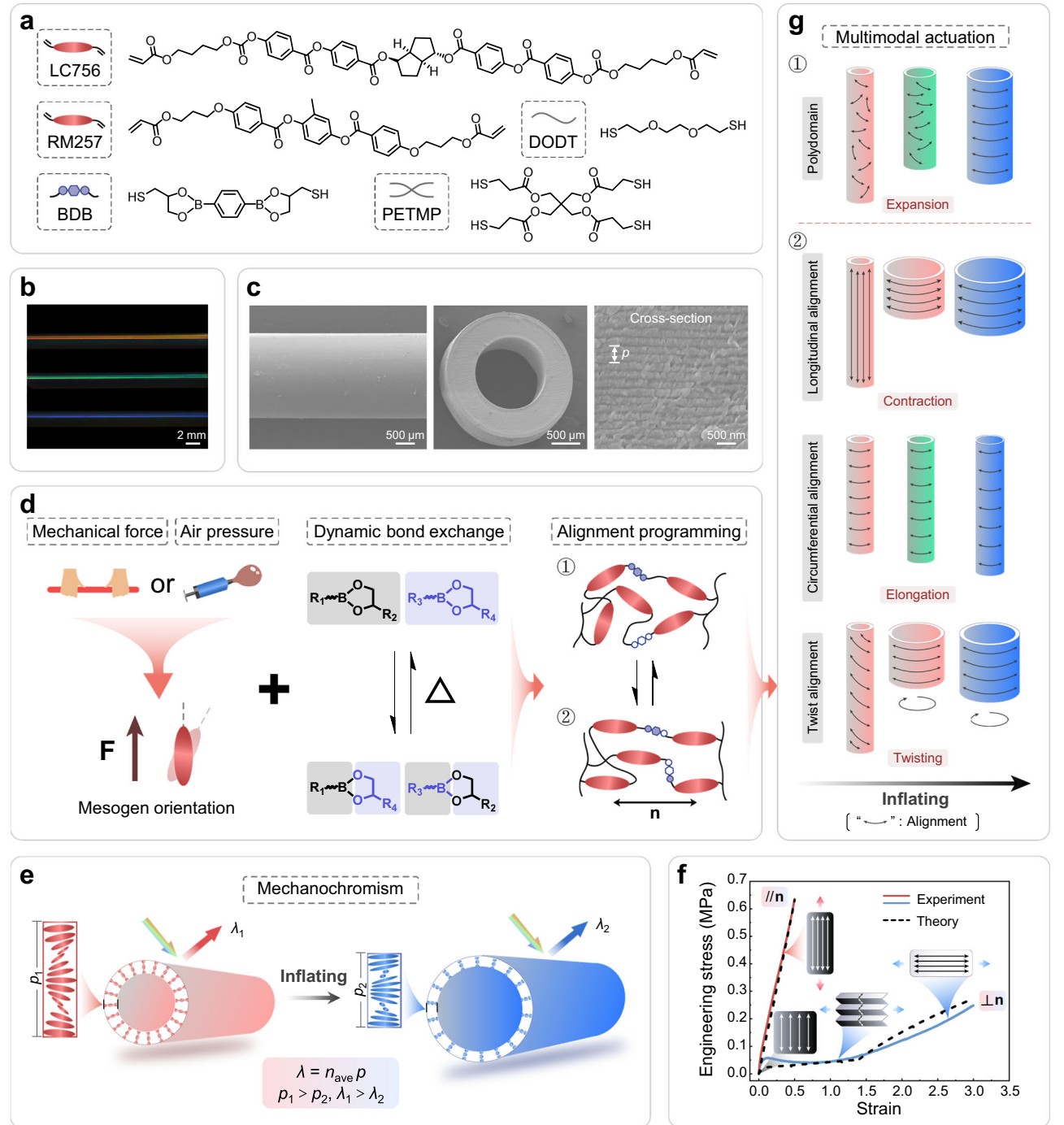

**Fig. 1 | Morphology and working principle of CLCE hollow fiber. a** The chemical composition of CLCE hollow fiber. **b** Photos of three CLCE hollow fibers that show red, green and blue. **c** SEM images of the longitudinal direction and cross-section of CLCE hollow fiber. **d** Alignment programming process of CLCE hollow fiber. **e** The color change mechanism of CLCE hollow fiber. $\lambda$ is the central wavelength of reflection. $n_{ave}$ is the average LC refractive index. The pitch ($p$) is defined as the length over which the LC mesogens rotate 360° along the helical axis. **f** The mechanical behavior of monodomain CLCEs under loads parallel and perpendicular to the alignment. The theoretical curve is obtained by choosing the polymer anisotropy $r = 5.5$, non-ideality $\alpha = 0.05$, and shear modulus $\mu_1 = 350$ kPa, $\mu_2 = 122.5$ kPa using the anisotropic model. See ref. 3 and the theory section for the mechanical model and notations. **g** The ensuing inflation deformation trajectory and color change of CLCE hollow fiber with different alignments under pneumatic actuation.

nematic directors in longitudinal, circumferential, and twisted geometries (Fig. 1d). During subsequent inflation of these hollow fibers, we observe color shifts, but also a rich range of motions, including expansion, contraction, elongation, and twisting (Fig. 1e, f, g). These phenomena can occur monotonically or non-monotonically and continuously via the coupling of director rotation and ballooning instabilities. To understand our observations, we develop a mechanical model of programmed cholesteric hollow fibers that is derived from the standard non-ideal energy of nematic elastomers, but with a suitable initial director profile that mimics the CLCE alignment after programming. This semi-analytical model successfully captures both highly non-trivial inflation mechanics of the CLCE hollow fibers and their color changes. Taken together, this study thus demonstrates that the rich soft-elastic behaviors of nematic LCEs can be combined with the vivid colors of CLCEs to create pneumatic actuators that both shift in color and undergo bespoke and complex deformations on inflation. We hope this provides a paradigm for future work on multi-responsive CLCEs, in which the sophisticated possibilities for mechanical design opened by director programming in nematic LCEs can be combined with CLCE colors to create multi-responsive systems in which both shape and color response can be designed, optimized, or even addressed independently, advancing the development of adaptive color-changing and intelligent actuation systems.

## Results

### Fabrication and characterization of CLCE hollow fiber

The CLCE hollow fiber was prepared via a template method combined with anisotropic deswelling. In a typical experiment, as shown in Fig. 2a, mold with a hollow ring structure was assembled using an inner capillary glass tube and an outer silicone tube. The CLCE precursor, containing RM257 as the mesogen, LC756 as the chiral agent, DODT and BDB as spacers, and PETMP as the crosslinker, was filled into the mold. Notably, BDB, which contains dynamic covalent B−O bonds enabling bond exchange, was synthesized following a reported procedure (Supplementary Fig. 1)[38]. When the molar content of BDB relative to the total spacers was $x$%, the sample was denoted as $x$BDB (20BDB was used in the experiment unless otherwise specified). Subsequently, a Michael addition reaction was conducted to form the first-step crosslinking network, accompanied by solvent evaporation. During this process, as the solvent mainly evaporated through the silicone tube, the LC molecules self-assembled into a periodic helical nanostructure[39]. Following this, a photo-crosslinking reaction under UV irradiation led to the formation of the second-step crosslinking network. After demolding, the CLCE hollow fiber was obtained. It is important to note that in the resulting CLCE hollow fiber, the helical arrangement of the LC mesogens was oriented radially (along the wall thickness direction), with the helical axis perpendicular to the fiber's longitudinal and circumferential directions. Along the longitudinal direction (for example, at the surface of the fiber), however, the LC mesogens exhibited a polydomain distribution without directional arrangement.

The color of the CLCE hollow fiber could be easily tuned by adjusting the content of the chiral agent. As shown in Fig. 1b and Supplementary Fig. 2, CLCE hollow fibers with red, green, and blue reflection colors were successfully obtained. The structure of the CLCE hollow fiber was characterized using scanning electron microscopy (SEM). As shown in Fig. 1c, the longitudinal section of the CLCE hollow fiber exhibited a uniform structure and a smooth surface. The cross-section revealed a concentric annular structure, with an inner diameter of ~1300 μm, an outer diameter of ~2200 μm, and a wall thickness of ~450 μm. At higher magnification in SEM, a typical lamellar pattern was observed, indicating helical alignment. To confirm the polydomain, polarized optical microscopy (POM) and X-ray diffraction (XRD) were performed. As shown in Fig. 2b, in the POM images, the CLCE hollow fiber showed no significant changes when rotated at 45° intervals,

consistently displaying a certain brightness. It was further confirmed by XRD in Fig. 2c, where the XRD pattern displayed a symmetric ring, and the azimuthal scan showed a flat line. These were consistent with the polydomain feature of the obtained CLCE hollow fiber.

In summary, by utilizing an anisotropic deswelling-assisted template method, we successfully fabricated polydomain CLCE hollow fibers with significant structural colors, laying the foundation for subsequent actuation studies.

## Pneumatic response performance of polydomain CLCE hollow fiber

The hollow structure of the CLCE hollow fiber facilitates its accommodation of air pressure. Since the polydomain CLCE hollow fiber was directly obtained through fabrication experiments, we first investigated its actuation behavior, selecting a red CLCE hollow fiber to allow for a larger visible color-changing range. As shown in Fig. 2d, one end of the CLCE hollow fiber was sealed and clamped, while the other end was connected to an air pump. Pneumatic actuation was carried out at 25 °C. The pressure difference between the inside and outside of the cavity was denoted as $\Delta p$. The initial fiber length is denoted as $L_0$, the fiber length at a stated pressure as $L$, and the length strain is defined as the ratio of $\Delta L/L_0$, where $\Delta L = L - L_0$. The initial outer fiber diameter is denoted as $D_0$, the outer fiber diameter at a stated pressure is referred to as $D$, and the outer diameter expansion ratio/strain is defined as $\Delta D/D_0$, where $\Delta D = D - D_0$. It was observed that as $\Delta p$ increased, the CLCE hollow fiber exhibited axial contraction and significant radial expansion. As illustrated in Fig. 2e, when $\Delta p$ reached 80 kPa, further inflation caused continued radial expansion but axial elongation. The maximum axial contraction ratio reached -17%, accompanied by a radial expansion ratio of ~44%. Regarding structural color, as shown in Fig. 2d, the color of the CLCE hollow fiber gradually underwent a blueshift as $\Delta p$ increased, transitioning from the initial red to green at 60 kPa, and then to blue at 100 kPa, with its reflection wavelength changing from 608 nm to 424 nm. Further inflation to 120 kPa caused the reflection spectrum to shift to even shorter wavelengths (Fig. 2f, g). The entire continuous pneumatic process was shown in Supplementary Movie 1. Moreover, it can be observed that as the pressure increased, the circular polarization selectivity of the polydomain CLCE hollow fiber disappeared (Supplementary Fig. 3 and Fig. 4), indicating the formation of directional alignment and distorted structures[11]. The mechanical response of the polydomain CLCE sample is given in Supplementary Fig. 5. Additionally, the polydomain CLCE hollow fiber maintained stable deformation and color-changing performance after 100 actuation cycles under 100 kPa pressure, demonstrating its excellent cyclic performance and laying the foundation for further applications (Fig. 2h).

To better understand this complex deformation trajectory theoretically, we develop a membrane balloon model for the CLCE that incorporates director rotation and its resultant (semi-)soft elasticity. We describe the CLCE as a cylindrical membrane with initial radius $R_0$, length $L_0$ and thickness $t_0$. Upon inflation, the balloon stretches $L_0 \to \lambda L_0$, dilates $R_0 \to \eta R_0$, and consequently thins $t_0 \to t_0/(\lambda\eta)$ to conserve the elastomer's volume. Working in $(z, \theta, \rho)$ cylindrical coordinates, the deformation gradient in the CLCE is thus simply $\mathbf{F} = \text{diag}(\lambda, \eta, 1/(\lambda\eta))$, which is constant throughout the membrane. To model the elastic response to these deformations, we then use the standard non-ideal neo-classical hyper-elastic energy density for a nematic elastomer fabricated with initial director $\mathbf{n}_0$, then subject to a deformation gradient $\mathbf{F}$ that rotates the alignment to the final state director $\mathbf{n}$:

$$W(\mathbf{F}, \mathbf{n}_0) = \min_{\mathbf{n}} \left[ \frac{1}{2}\mu \text{Tr}\left( \mathbf{F}^T \ell^{-1}(\mathbf{n}) \mathbf{F} \ell(\mathbf{n}_0) + \alpha \mathbf{F}(\mathbf{I} - \mathbf{n}_0 \mathbf{n}_0) \mathbf{F}^T \mathbf{n}\mathbf{n} \right) \right],$$

$$\text{where } \ell(\mathbf{n}) = r^{-1/3}(\mathbf{I} + (r-1)\mathbf{n}\mathbf{n}). \tag{1}$$

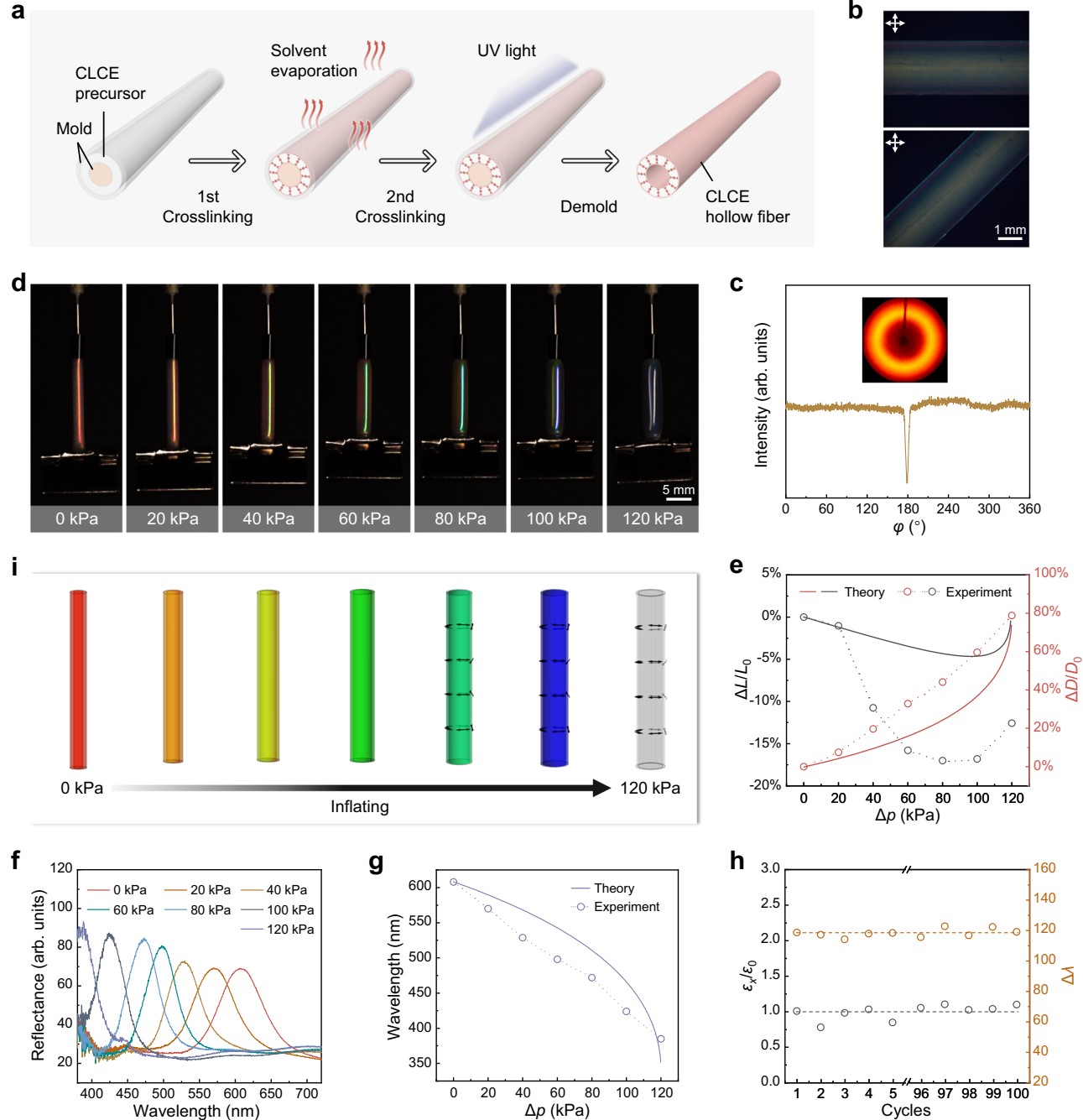

**Fig. 2 | Characterization of pneumatic response performance of polydomain CLCE hollow fiber. a** Schematic of fabrication process for CLCE hollow fiber via an anisotropic deswelling-assisted template method. **b** POM of polydomain CLCE hollow fiber viewed between crossed polarizers. The fiber-axes were aligned parallel to and at 45° to the polarizer, respectively. **c** XRD pattern and intensity of azimuthal scan of polydomain CLCE hollow fiber. **d** Images of a red polydomain CLCE hollow fiber inflated at different pressures. **e** Axial and radial deformation ratios of the polydomain CLCE hollow fiber as a function of pressure. **f** Reflectance spectra of the CLCE hollow fiber corresponding to **d. g** The reflection wavelength of the polydomain CLCE hollow fiber at different pressures. **h** Cyclic performance of deformation and color change of the polydomain CLCE hollow fiber under pneumatic actuation at 100 kPa. The length strain of the first cycle is denoted as $\varepsilon_0$, and the length strain of the $x$-th cycle is denoted as $\varepsilon_x$. The stability of the contraction ratio is characterized by $\varepsilon_x/\varepsilon_0$. The difference in reflection wavelength before and after inflation is denoted as $\Delta\lambda$. **i** Theoretical simulation results of the mesogen director field, deformation, and color change for the polydomain CLCE hollow fiber under different pressures.

The first term above describes an ideal nematic elastomer with shear modulus $\mu$ and polymer anisotropy $r > 1$. This term alone exhibits perfect soft elasticity, in which "soft modes" of deformation of the general form $\mathbf{F} = \boldsymbol{\ell}^{1/2}(\mathbf{n})\,\boldsymbol{\ell}^{-1/2}(\mathbf{n}_0)$ can be accommodated by director rotation but without any increase in energy. The second "non-ideal" term encodes a bias for $\mathbf{n}$ to align with its fabrication direction, $\mathbf{n}_0$, causing these soft-modes to incur an energy proportional to the

(small) coefficient of non-ideality, $\alpha$. In our CLCEs, the initial director is planar, $\mathbf{n}_0 = (\cos\theta, \sin\theta, 0)$, but twists through the thickness with helical pitch $p$, giving $\theta(\rho) = 2\pi\rho/p$, so the total elastic energy can be written as a thickness average $E_{el} = \pi(R_{\text{out}}^2 - R_{\text{in}}^2)\,L_0 < W(\mathbf{F}, \mathbf{n}_0)>_\rho$ with inner radius $R_{\text{in}}$ and outer radius $R_{\text{out}}$. Finally, at a given pressure $P$, the deformation of the balloon is given by minimizing $E_{el} - PV$, ($V$ being internal volume) over $\lambda$ and $\eta$.

Conducting the thickness average and minimization numerically in Mathematica (see Theoretical supplement), using the measured balloon geometry, and the three material parameters obtained independently by fitting to monodomain stress-strain curves (Fig. 1f), we obtain a theoretical prediction for the shifting shape and color of the hollow fiber (Fig. 2i). Though agreement is not perfect, the theory captures the key surprise, which is that the balloon initially shortens then lengthens, while the color is blueshifted monotonically. To rationalize this, we consider that, as ever in cylindrical balloons, the hoop stress is twice the longitudinal stress. In a region with initial longitudinal alignment, this difference induces the director to rotate toward the circumferential direction, and, in isolation, in an ideal system, this would occur via a perfect soft mode $\mathbf{F} = \mathrm{diag}(r^{1/2}, r^{1/2}, 1)$ that causes radial dilation and longitudinal contraction; after the director reaches circumferential, no further rotation is available, so the response is like a normal rubber balloon, with longitudinal and radial expansion. The behavior of the CLCE balloon, with modest but non-monotonic longitudinal contraction, arises as a compromise between the initially more longitudinal regions and more circumferential regions within each helix, which must all deform in the same way. Inspired by the developed theory, these observed experimental behaviors could be attributed to the fact that, although the polydomain CLCE hollow fiber exhibited overall mechanical isotropy, its internal stress distribution under inflation became anisotropic, with the hollow fiber structure experiencing doubled circumferential (hoop) stress compared to longitudinal stress under the same pressure. This anisotropy subsequently led to director reorientation toward the circumferential direction for part of the director domain, resulting in dramatic longitudinal contraction. In particular, the longitudinal strain exabits a plateau below 20 kPa, likely owing to the semi-soft behavior and the degree of alignment. When the pressure reached around 80 kPa, the director reorientation was saturated and all the directors were circumferential. This is consistent with the result that axial contraction reached its maximum at this pressure. After that, the fiber got thinner by further inflating, resulting in the shortening of the helical pitch because of the volume conservation during inflation.

Combining all the experiments and theoretical simulations in Fig. 2, it can be concluded that the polydomain CLCE hollow fiber can simultaneously undergo deformation and color changes under pneumatic actuation, with the magnitude of these changes adjustable by pressure. During the experimental actuation process, the fiber exhibited maximum values of -17% length strain, -79% radial expansion, and a 223 nm blueshift in reflection wavelength. The theory captures the key non-monotonic surprise of the experiment, which is attributed to director-rotation towards circumferential. Unlike in our later case, the quantitative agreement here is somewhat disappointing, which we attribute to our treatment of neighboring domains as independent, neglecting the complex micro-mechanics that can arise from cooperation[40]. Also, our unified theoretical model performs better in the semi-soft mode regime, whereas the deformation of a polydomain sample is partly in the "hard" mode, with no director rotation, since the polydomain contains fully helical directors. Nevertheless, the model already captures several key features, including a non-monotonic and small length change, a monotonic and large diameter change, and a large monotonic blueshift.

### Pneumatic response performance of longitudinally aligned CLCE hollow fiber

The intrinsic mechanical anisotropy and (semi-)soft elasticity endow LCE materials with diverse pneumatic actuation motions[29,30,34,35,41], which also holds potential for CLCE systems. To achieve this functionality, it is necessary to impart directional arrangements to CLCEs. However, the CLCE hollow fiber obtained through the aforementioned experiments lacked an overall orientation. To address this issue, we introduced boronic ester molecules into the chemical system, aiming

to program an overall alignment of the CLCE hollow fiber through thermo-activated exchange of dynamic covalent B−O bonds. The effect of dynamic B−O bonds on the performance of the CLCE hollow fiber was investigated by varying the content of BDB. Differential scanning calorimetry (DSC) curves indicated that the LC-to-isotropic transition temperature ($T_{ni}$) of the CLCEs with different BDB contents was approximately 84.5 °C (Supplementary Fig. 6). Stress relaxation tests of the CLCEs showed that the 20BDB sample relaxed approximately 80% of the stress within 60 minutes, while the 50BDB sample achieved complete stress relaxation (Fig. 3b), demonstrating the thermo-activated exchange capability of the dynamic covalent B−O bonds.

To obtain longitudinally aligned CLCE hollow fibers, the polydomain CLCE hollow fiber was stretched to twice its original length and maintained at 65 °C (20 °C below $T_{ni}$) for 6 h. Due to the thermo-activated B−O bond exchange, the polymer network of the CLCE hollow fiber formed a new topological structure to adapt to the aligned state. After cooling to room temperature and removing the fiber from the clamp, longitudinally aligned CLCE hollow fibers were obtained (Fig. 3a). Notably, since stretching reduces the wall thickness of the hollow fiber and causes a blueshift in the reflection color, polydomain fibers with an initial reflection wavelength in the near-infrared (NIR) range were selected to ensure that the resulting longitudinally aligned CLCE hollow fibers exhibited a red color. As shown in Supplementary Fig. 7, in this experiment, the reflection wavelength of the CLCE hollow fiber before longitudinal alignment programming was 920 nm, which shifted to 682 nm after programming. The alignment state of the CLCE hollow fiber before and after programming was investigated using POM and XRD. As shown in Fig. 3c and Supplementary Fig. 8, the CLCE hollow fiber was bright under POM before programming, with no significant changes when rotated at 45° intervals. In contrast, after programming, the sample appeared completely dark when its long axis was parallel to one of the polarizers, and reached maximum brightness when rotated 45° relative to the polarizer, indicating that the programmed CLCE hollow fiber achieved almost homogeneous longitudinal alignment, while maintaining enough residual wobble alignment to preserve the structural color (same for other programmed alignments). As shown in Fig. 3d, the XRD patterns changed from a ring to a pair of diffraction arcs, consistent with the fact that the CLCE hollow fiber transitioned from a polydomain to an aligned monodomain structure after programming. The order parameter $f$ of the longitudinally aligned CLCE hollow fiber was 0.63, as determined by the Hermans−Stein orientation distribution function[42–44], indicating good alignment along the long axis. The above results demonstrated that a red, longitudinally aligned CLCE hollow fiber can be obtained through programming using thermo-activated B−O bond exchange.

Next, pneumatic actuation was performed on the longitudinally aligned CLCE hollow fiber. As shown in Fig. 3e, when $\Delta p$ reached 60 kPa, the CLCE hollow fiber exhibited almost no deformation. When $\Delta p$ reached 80 kPa, a clear and sudden expansion occurred, accompanied by substantial axial contraction, indicating a sub-critical instability. This sudden deformation contrasted with the gradual deformation observed in the polydomain CLCE hollow fiber. The critical pressure $\Delta p$ at which the CLCE hollow fiber began to undergo abrupt deformation was defined as $\Delta p_c$. Based on these observations, it can be concluded that the instability occurs between 60 kPa and 80 kPa. It was also observed that when $\Delta p$ exceeded $\Delta p_c$, the CLCE hollow fiber began to exhibit slight reverse elongation along with further radial expansion. Throughout the process, the maximum axial contraction ratio of the longitudinally aligned CLCE hollow fiber was -50%, with a corresponding radial expansion ratio of -130% (Fig. 3f). In terms of color change, as shown in Fig. 3e, the CLCE hollow fiber remained red as $\Delta p$ increased, with a slight blueshift in its reflection spectrum (Fig. 3g, h). Although a drastic contraction occurred at 80 kPa, the reflection wavelength shifted only from 655 nm to 605 nm

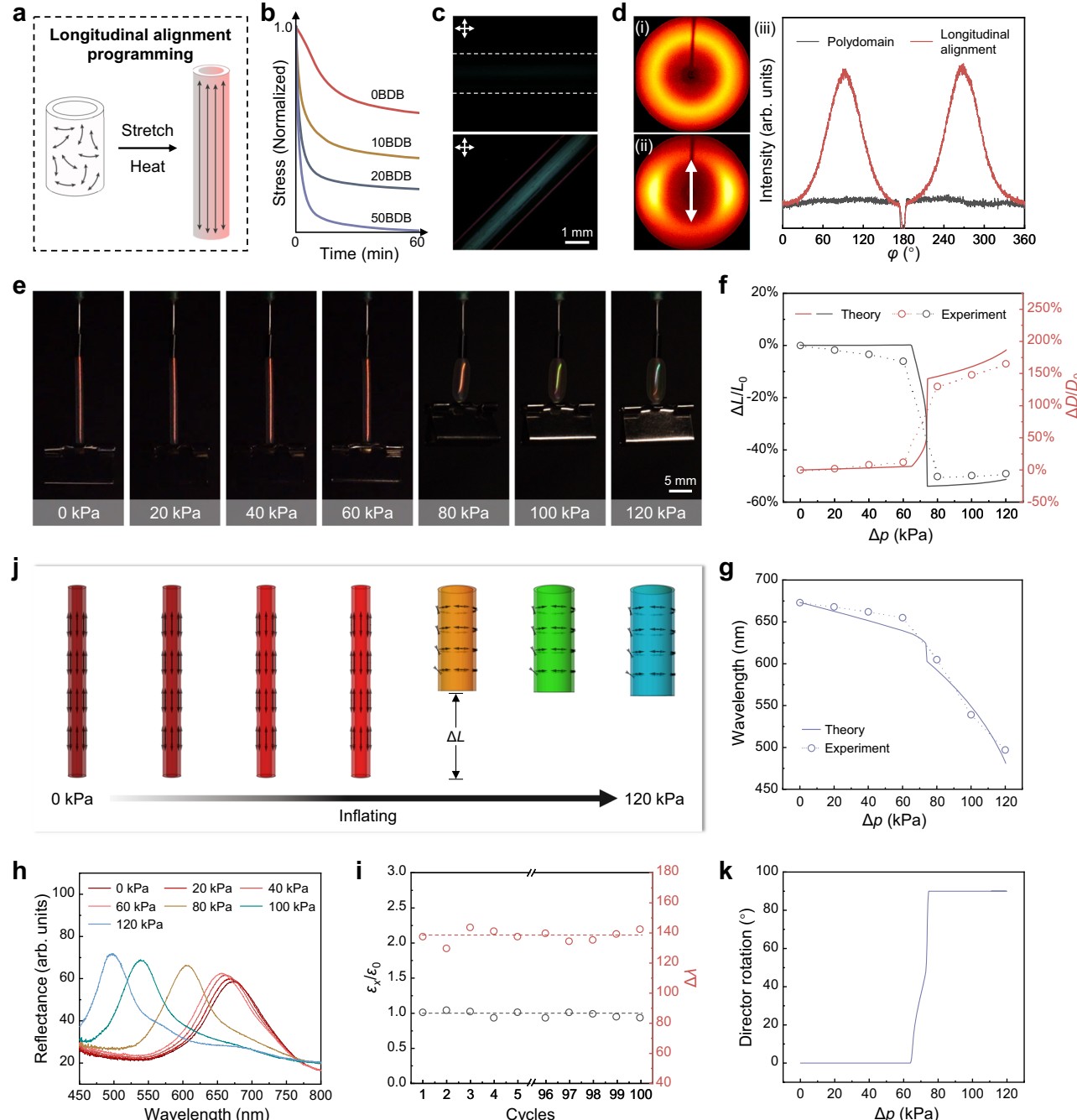

**Fig. 3 | Characterization of pneumatic response performance of longitudinally aligned CLCE hollow fiber. a** Schematic diagram of the programming process for obtaining CLCE hollow fiber with longitudinal alignment. **b** Stress relaxation behavior of CLCEs with 0, 10, 20, and 50 BDB content at 65 °C. **c** POM images of CLCE hollow fibers with longitudinal alignment. **d** XRD patterns (left) and intensity of azimuthal scan (right) of CLCE hollow fiber before and after programming with longitudinal alignment. The arrow indicates the orientation direction. **e** Images of a red CLCE hollow fiber with longitudinal alignment inflated at different pressures. **f** Axial and radial deformation ratios of the CLCE hollow fiber with longitudinal alignment as a function of pressure. **g** The reflection wavelength of the longitudinally oriented CLCE hollow fiber at different pressures. **h** Reflectance spectra of the CLCE hollow fiber corresponding to **e**. **i** Cyclic performance of deformation and color change of longitudinally aligned CLCE hollow fiber under pneumatic actuation at 120 kPa. **j** Simulation results of the mesogen director field, deformation, and color change for the CLCE hollow fiber with longitudinal alignment under different pressures. **k** Director rotation angle as a function of pressure.

and remained within the red region. This is because, in the ideal case, pure director rotation does not alter either the fiber thickness or the photonic reflection band. Once the directors had fully rotated into the circumferential direction, further increases in pressure induced the thickness thinning, causing the reflection color of the CLCE hollow fiber to gradually shift from red to blue (from 605 nm to 497 nm). These results demonstrated that although the CLCE hollow fiber underwent large deformation at $\Delta p_c$, its wall thickness did not decrease substantially. The entire continuous pneumatic process was shown in Supplementary Movie 2. Moreover, it can be observed that the CLCE hollow fiber exhibited no circular polarization selectivity— neither in its initial longitudinally aligned state nor after reaching the $\Delta p_c$ and transitioning to a circumferential alignment—indicating that the aligned samples possess a distorted structure (Supplementary

Figs. 9 and Fig. 10). Additionally, 100 pneumatic actuation cycles at 120 kPa confirmed the actuation stability of the CLCE hollow fiber, demonstrating its potential as a dual-deformation and color-changing component for future applications (Fig. 3i).

We notice that several previous studies have examined how soft-elasticity and inflation are expected to combine in cylindrical nematic balloons with initially longitudinal alignment[29–32]. The key mechanics are that, since inflation always generates a hoop stress that is twice the longitudinal stress, the balloon initially inflates via a soft-mode, in which the director rotates to the circumferential direction. As observed experimentally[30], the balloon actually gets shorter. Furthermore, during director rotation, the soft elastic modes of deformation demand shears that must twist the balloon. In strictly longitudinal balloons, rotation and twist are equally likely to occur in either sense, likely leading them to cancel out macroscopically[29,31]. However, these phenomena have not been analyzed in CLCEs.

To quantitatively understand the experiments, we use the same energy form but with the anchored director pointing towards the axial direction superposed by a small oscillation, $\theta(\rho) = 0.05 \sin(2\pi\rho/p)$. The oscillation is caused by the incompatibility of the director rotation through the thickness during the mechanical programming, yielding the programmed director pointing almost but not perfectly along the longitudinal direction through the thickness, and allowing the structural color to survive programming. Again, the theoretical model is solved numerically for the pressure-controlled ballooning problem. As shown in Fig. 3j, using the same material parameters, our model successfully captured the main mechanical response. In particular, both the length and the diameter have a large jump at a critical pressure, with length shortening and diameter increasing. Meanwhile, the thickness change is also calculated which predicts the color change accurately (see the theoretical manifestations in Fig. 3j), showing that the blueshift largely arises above the critical pressure. As previewed in the polydomain case, an ideal ($\alpha = 0$) longitudinal balloon would jump to a circumferential alignment at zero pressure via a soft mode, $\mathbf{F} = \text{diag}(r^{1/2}, r^{1/2}, 1)$, with radial dilation, longitudinal expansion, and interior volume increasing by $r^{1/2}$. In CLCEs, this transition is delayed by non-ideality and the initial director fluctuation, but as anticipated in previous work[29], it remains a subcritical jump. To confirm this mechanism, we plot the director rotation implied by our theory versus pressure in Fig. 3k. We see that the director rotation and the jump emerge simultaneously, confirming the jump is associated with the soft mode, and thus fundamentally different from a usual neo-Hookean balloon. Taken together, the experimental results and theoretical simulations demonstrated that the longitudinally aligned CLCE hollow fiber responded to air pressure, achieving maximum values of ~50% axial contraction, ~165% radial expansion, and a blueshift of up to 176 nm. Inspired by the theory, this anomalous inflation behavior was primarily attributed to the simultaneous director reorientation and ballooning instability, under the dominant hoop stress[5,45]. This caused the circumferential direction to deform more easily during inflation, resulting in radial expansion and simultaneous axial contraction. In addition, due to the semi-soft elasticity of CLCEs, the LC mesogens of the CLCE hollow fiber exhibited almost no rotation below $\Delta p_c$[46–49]. Upon reaching $\Delta p_c$, a slight increase in $\Delta p$ induced significant rotation of the LC mesogens. The simultaneous rotation of all mesogens from the longitudinal to the circumferential direction macroscopically manifested as a sudden axial contraction of the CLCE hollow fiber. Moreover, when the pressure was increased beyond $\Delta p_c$, the mesogen rotation was saturated and this led to further axial elongation and radial expansion as normal balloon.

From the results shown in Fig. 3, it can be concluded that the CLCE system containing B−O bonds exhibits effective dynamic bond exchange capability. Thermo-activated programming of polydomain CLCE hollow fibers can yield longitudinally aligned CLCE hollow fibers. The simultaneous shape and color changes of the longitudinally

aligned CLCE hollow fiber under pneumatic actuation are attributed to its mechanical anisotropy and soft elasticity, during which the LC mesogens rotate from longitudinal to circumferential alignment. Notably, the distinct "plateau-then-jump" response presents opportunities for designing functional devices, such as threshold-type optical sensors and multi-stage soft actuators. By leveraging the pneumatic actuation-induced adaptability alongside the dual deformation and color-changing capabilities, the practical application scope of CLCE materials is expected to be further expanded.

## Pneumatic response performance of circumferentially aligned CLCE hollow fiber

Different directional programming of LC mesogens may lead to distinct actuation behaviors. Longitudinally aligned CLCE hollow fibers can be obtained through simple axial stretching and polymer network rearrangement. However, achieving circumferential alignment in fibrous materials by applying circumferential force remains a challenge. Surprisingly, pneumatic actuation, as a method of applying mechanical force, provided an opportunity to obtain circumferentially aligned CLCE hollow fibers. As mentioned in the previous section, longitudinally aligned CLCE hollow fibers undergo a transition where the LC mesogens rotate from the axial to the circumferential direction upon reaching $\Delta p_c$. If this oriented state is fixed, circumferentially aligned CLCE hollow fibers can be obtained. As shown in Fig. 4a, circumferential alignment programming of the CLCE hollow fiber was achieved by combining pneumatic actuation and dynamic covalent B−O bond exchange. Dynamic mechanical analysis (DMA) revealed that the Young's modulus of the monodomain CLCEs decreased with increasing temperature (Fig. 4b)[35,50]. Therefore, in the experiment, circumferential alignment of the CLCE hollow fiber was programmed at 65 °C under a lower pressure of 70 kPa. Heating for 6 h allowed sufficient rearrangement of the topological network, resulting in circumferentially aligned CLCE hollow fibers. Notably, since the CLCE hollow fiber was inflated around $\Delta p_c$, its color remained in the red state. Thus, samples with reflection wavelengths in the red range were selected for programming, and the resulting circumferentially aligned CLCE hollow fibers also exhibited a red color. To verify the orientation state, POM and XRD tests were conducted. Figure 4c shows that when the long axis of the CLCE hollow fiber was parallel to one polarizer, the POM image appeared dark, whereas when the long axis was at a 45° angle to the polarizer, the POM image appeared bright, indicating a directional arrangement. The XRD pattern displayed a pair of distinct diffraction arcs, with the orientation direction perpendicular to the axial direction of the CLCE hollow fiber (The long axis of the fiber sample during testing was perpendicular to the beam stopper). The order parameter $f$ was calculated to be 0.60, indicating that the circumferentially aligned CLCE hollow fiber was well-oriented (Fig. 4d). These experimental results demonstrate that by inflating the longitudinally aligned CLCE hollow fiber to rotate the LC mesogens into a circumferential state, while heating to induce bond exchange reactions, red circumferentially aligned CLCE hollow fibers were successfully obtained.

Under pneumatic actuation, as shown in Fig. 4e and Supplementary Movie 3, the CLCE hollow fiber exhibited a gradual axial elongation accompanied by radial expansion as the inflation pressure increased. When the pressure reached 200 kPa, the maximum axial elongation ratio was ~14%, and the maximum radial expansion ratio was ~25% (Fig. 4f). The color gradually changed from red to blue, with the corresponding reflection spectrum shifting from 611 nm to 453 nm, showing an approximately monotonic linear relationship with the change in pressure (Fig. 4g, h). This behavior is consistent with the pneumatic response of longitudinally aligned CLCE hollow fibers after reaching $\Delta p_c$, as shown in Fig. 3. Additionally, the circumferentially aligned CLCE hollow fiber exhibited excellent stability over 100 cycles of pneumatic actuation (Fig. 4i), demonstrating its reliability for practical applications.

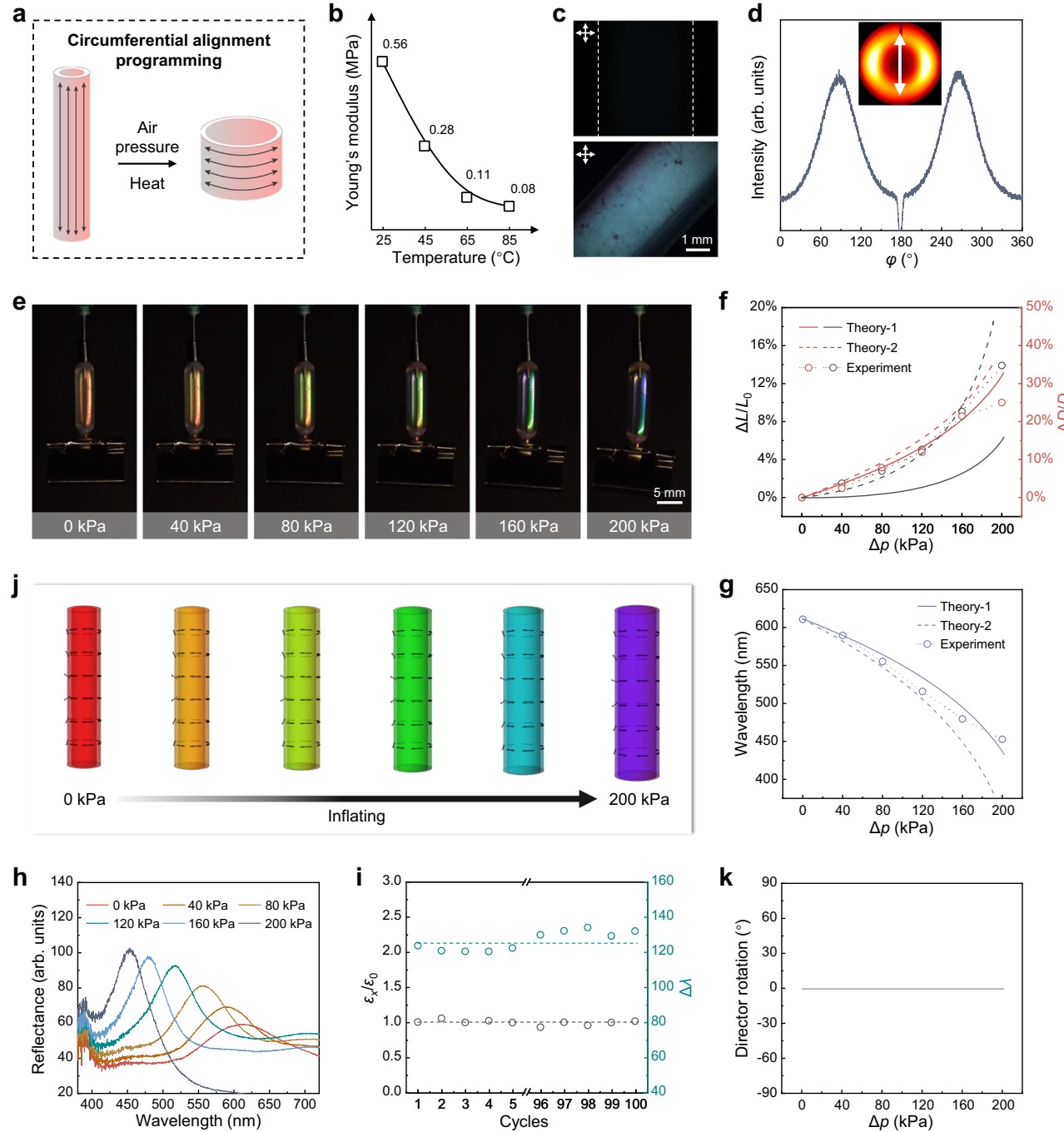

**Fig. 4 | Characterization of pneumatic response performance of circumferentially aligned CLCE hollow fiber. a** Schematic diagram of the programming process for obtaining CLCE hollow fiber with circumferential alignment. **b** The Young's modulus of the monodomain CLCEs at different temperatures. During the test, the load direction was perpendicular to the alignment. **c** POM images of CLCE hollow fibers with circumferential alignment. **d** XRD pattern and azimuthal scan intensity of the CLCE hollow fiber with circumferential alignment. To avoid the influence of the beam stopper on the results, the circumferentially aligned sample was placed horizontally during testing. Samples with other alignment modes were placed vertically relative to the instrument, at a 90° difference. As a result, the diffraction arc direction of the circumferential sample in the XRD pattern was consistent with that of, for example, the longitudinally aligned sample. **e** Images of a red CLCE hollow fiber with circumferential alignment inflated at different pressures. **f** Axial and radial deformation ratios of the CLCE hollow fiber with circumferential alignment as a function of pressure. **g** The reflection wavelength of the circumferentially oriented CLCE hollow fiber at varying pressures. **h** Reflectance spectra of the CLCE hollow fiber corresponding to **e**. **i** Cyclic performance of deformation and color change of circumferentially aligned CLCE hollow fiber under pneumatic actuation at 160 kPa. **j** Simulation results of the mesogen director field, deformation, and color change for the CLCE hollow fiber with circumferential alignment under different pressures. **k** Director rotation angle as a function of pressure.

The actuation is theoretically predicated by the same method, with the programmed anchored director pointing circumferentially superposed by a small oscillation: $\theta(\rho) = 0.5\pi + 0.05 \sin(2\pi\rho/p)$. As shown in Fig. 4j, the main features of the length change, diameter change, and color change are captured by the theory. In contrast with the longitudinal alignment case, the length and diameter changes exhibit no jumps, because the director is already circumferential which gives no room for further director reorientation (Fig. 4k). However, the

length change predicted by our model is noticeably smaller than that observed in experiments. We attribute this discrepancy to the isotropic modulus assumed in our original neo-classic energy model, whereas in reality the modulus is anisotropic along and perpendicular to the director. To address this, we also employ a modified model (Theory-2 in Fig. 4f) that incorporates soft elasticity and modulus anisotropy[51], which yields improved predictions (see the theoretical supplement for details). Nevertheless, we primarily retain the original model, as it already captures the main features with satisfactory agreement.

From the results shown in Fig. 4, it can be concluded that the method of applying force through inflation enables the formation of circumferentially aligned CLCE hollow fibers, and the oriented state can be fixed using thermal exchange of dynamic B−O bonds. The circumferentially aligned CLCE hollow fibers can respond to pneumatic actuation, producing simultaneous axial elongation of ~14%, radial expansion of ~25%, and a blueshift of 158 nm in color under a pressure of 200 kPa. During pneumatic actuation, circumferentially aligned CLCE hollow fibers exhibit both elongation and expansion, achieving a large wavelength shift within a small strain range, which shows great potential for applications in strain sensors, information displays, and other fields.

## Pneumatic response performance of twisted CLCE hollow fiber

By controlling the direction of the alignment, the mechanical anisotropy of CLCE hollow fibers could be programmed, thereby generating distinct pneumatic actuation motions and performances, such as the contraction of longitudinally aligned CLCE hollow fibers and the elongation of circumferentially aligned CLCE hollow fibers. However, the aforementioned two alignments are orthogonal to the long axis of the fiber. If the alignment of mesogens in CLCEs forms a certain angle with the long axis of the fiber, i.e., a twisted alignment, what kind of pneumatic actuation behavior would occur? We conducted an investigation into this matter. First, to achieve twisted alignment, during the programming process, we simultaneously stretched and twisted the polydomain CLCE hollow fiber, causing the LC mesogens to rotate at a certain angle along the long axis of the fiber (Fig. 5a). Heating was then applied to activate dynamic B−O bond exchange to fix the alignment. The degree of twisting was quantified by the twist density $\tau_0$, which is given by: $\tau_0 = \Delta\theta/L_0$, where $\Delta\theta$ is the torsional angle introduced into the CLCE hollow fiber and $L_0$ is the length of the CLCE hollow fiber. Through thermal programming, we successfully fabricated CLCE hollow fibers with $\tau_0$ of 9° mm$^{-1}$ and 18° mm$^{-1}$. It should be noted that the hollow structure would flatten if $\tau_0$ exceeded 18° mm$^{-1}$, which was the maximum value investigated in this study. Additionally, twisting in clockwise and counterclockwise directions resulted in opposite chirality. Notably, to ensure the twisted CLCE hollow fiber exhibited a red color, we selected samples with reflection spectra in the NIR range for programming. Figure 5b shows POM images of the left-handed twisted CLCE hollow fiber with $\tau_0$ of 9° mm$^{-1}$ and 18° mm$^{-1}$, respectively. It can be observed that the surface of the CLCE hollow fiber exhibited twisted textures, which can be attributed to the torsional stretching during preparation. The bias angle $\theta$ between the alignment direction of the twisted textures and the long fiber axis can be calculated as: $\theta = \tan^{-1}(R_{out}\tau_0)$, where $R_{out}$ is the outer radius of the CLCE hollow fiber. The theoretical bias angles calculated for the CLCE hollow fibers in Fig. 5b were 7.6° and 15.2°, which closely matched the experimentally measured average values of 6.4° and 14.7°. Moreover, when the long axis was parallel to one of the crossed polarizers, the POM image of the CLCE hollow fiber appeared bright, and the transmitted light intensity reached its maximum when the long axis was at a 45° angle to the polarizer. The right-handed twisted CLCE hollow fiber with $\tau_0$ of 9° mm$^{-1}$ and 18° mm$^{-1}$ exhibited the same phenomenon (Supplementary Fig. 11). This differed from the previously investigated polydomain CLCEs, which exhibited birefringence at all angles, as well as from the

longitudinally and circumferentially aligned CLCE hollow fibers, which both appeared completely dark when the long axis of the fiber was parallel to one of the polarizers. These results indicate that the mesogens in the twisted CLCE hollow fiber possessed a twisted alignment along the long axis. The twisted alignment of mesogens within the left-handed CLCE hollow fiber with $\tau_0$ of 9° mm$^{-1}$ and 18° mm$^{-1}$ was further confirmed by XRD, as shown in Fig. 5c. Compared to uniaxial alignment, the spot pattern gradually transformed into a wider arc as $\tau_0$ increased in twisted CLCE hollow fibers. The order parameter of the CLCE hollow fiber decreased to 0.61 and 0.58 when the twist density $\tau_0$ increased to 9° mm$^{-1}$ and 18° mm$^{-1}$, respectively. The reduction in the degree of orientation along the long axis of the fiber may be attributed to the alignment of mesogens along the torsional direction, deviating from the uniaxial alignment[31,32,52]. From the results above, it can be concluded that a CLCE hollow fiber can be programmed with a designed twisted alignment by controlling the twist density and chirality.

We further investigated the pneumatic actuation behavior of twisted CLCE hollow fibers. Figure 5d shows images of the left-handed CLCE hollow fiber with $\tau_0$ of 18° mm$^{-1}$ during inflation. It can be observed that, as $\Delta p$ increased, the CLCE hollow fiber demonstrated rotation, which became more pronounced due to the rotation of the clamp. The rotation angle $\phi$ was calculated as the angle between the clamp axis (black dashed line) and its initial position (gray dashed line) at different pressures Counterclockwise angles are recorded as negative (−), while clockwise angles are recorded as positive (+). Combining Fig. 5d and Supplementary Movie 4, when the pressure ranged from 0 to 60 kPa, the left-handed CLCE hollow fiber rotated counterclockwise (from the top view). When the pressure increased to 80 kPa, the fiber rotated counterclockwise to its maximum angle and then jumped backwards in rotation. After stabilization, further inflation caused the CLCE hollow fiber to rotate clockwise. During this process, the fiber rotated counterclockwise to a maximum angle of approximately 163°, then rotated clockwise by approximately 330° to +167° at 80 kPa, and reached a maximum rotation angle of approximately +185° at 140 kPa (Fig. 5e). Simultaneously, under pneumatic actuation, the CLCE hollow fiber also exhibited axial contraction and radial expansion, as shown in Fig. 5f. The magnitude of length contraction significantly increased at 80 kPa ($\Delta p_c$), with a maximum contraction ratio of ~48%. Beyond this point, as $\Delta p$ increased, the fiber exhibited axial elongation and radial expansion, similar to the deformation trend of longitudinally aligned CLCE hollow fibers. This rotation and deformation behavior can be attributed to the uniaxial and helical orientation of mesogens along the twisted CLCE hollow fiber. During inflation, the internal pressure increased, and the circumferential expansion force caused the mesogens to deflect toward the circumferential direction, resulting in length contraction and torsional deformation.

The left-handed CLCE hollow fiber with $\tau_0$ of 9° mm$^{-1}$ was also investigated (Fig. 5e and Supplementary Fig. 12). It exhibited counterclockwise rotation and axial contraction before reaching 80 kPa, rotating to a maximum angle of approximately 213° at 80 kPa before reversing its rotation by approximately 206° to −7°, with a maximum contraction ratio of ~48%. Beyond 80 kPa, the CLCE hollow fiber continued to rotate counterclockwise but exhibited axial elongation. Compared to the left-handed sample, the right-handed CLCE hollow fiber exhibited opposite rotation directions but similar rotation angles and deformation trends under pneumatic actuation. In terms of color, as shown in Fig. 5g, h, the reflection wavelength of the twisted CLCE hollow fiber gradually shifted as the pressure increased. After reaching 80 kPa, due to simultaneous radial expansion and axial elongation, the pitch length decreased more significantly, increasing the color-changing sensitivity. To demonstrate the reliability of the pneumatic actuation of twisted CLCE hollow fibers, as shown in Fig. 5i, we conducted 100 cycles of inflation at 140 kPa, laying a foundation for its further applications.

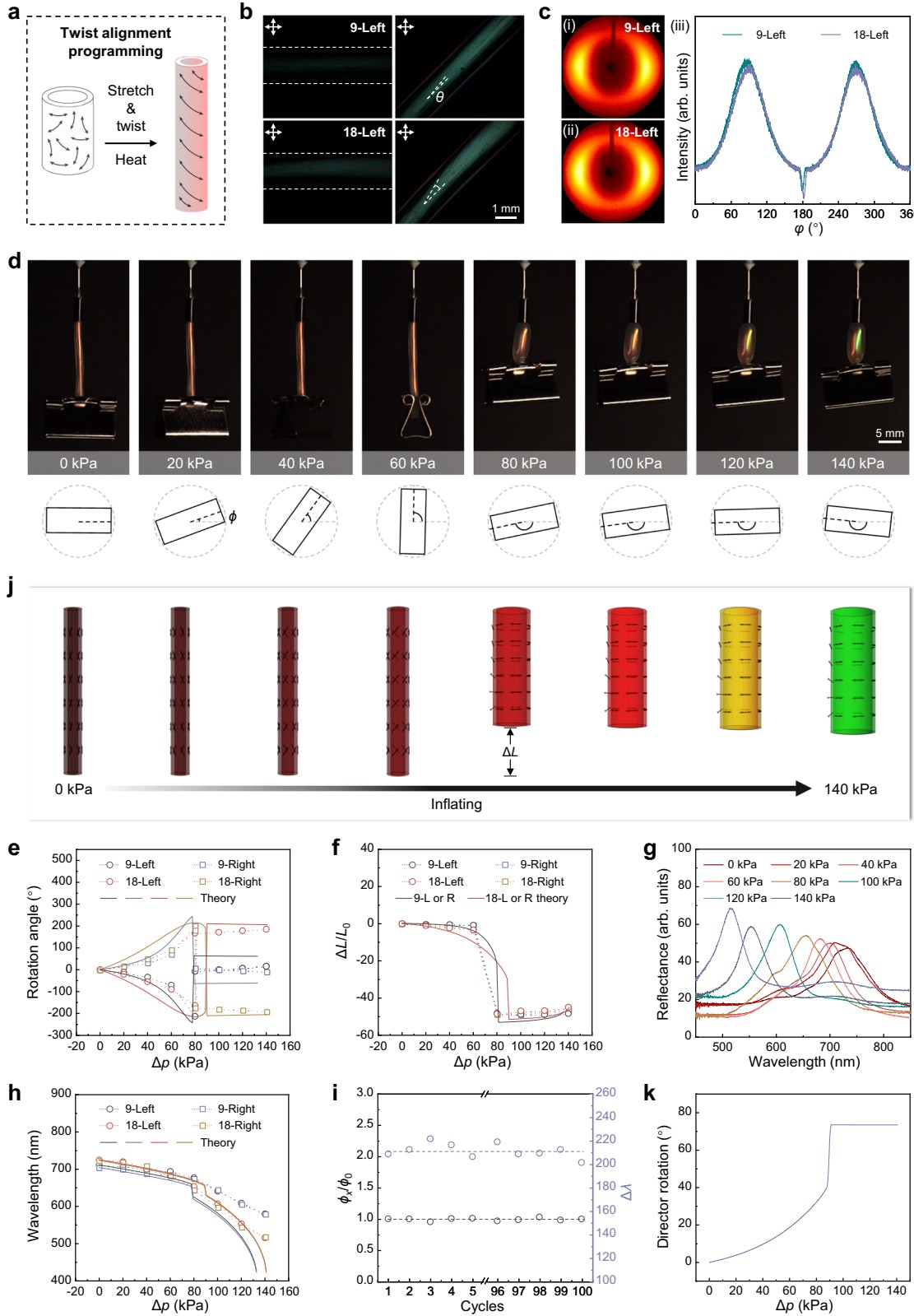

To rationalize this rich twisting behavior, we again used our mechanical model, now with a director profile $\theta(\rho) = \theta_0 + 0.05 \sin(2\pi\rho/p)$, that describes a small residual thickness oscillation superimposed on the main director making fixed angle $\theta_0$ with longitudinal. Importantly, in this case, the balloon deformation is also allowed to include a shear component, $\mathbf{F}_{\theta z} = \gamma$, which is included in the minimization and, when non-zero, describes the pneumatic

twisting of the balloon. In contrast with the longitudinal balloons, the initial bias angle in the helical balloons drives the choice of a single sign of $\gamma$, meaning that the shears turn into the macroscopic twists that we observe[32], enabling another mode of deformation. In the theoretical prediction, we choose the twist density $\tau_0$ close to the experimental values. The main direction $\theta_0$ of the anchored director can be calculated according to $\tau_0$. The geometric parameters $R_{out}$, $R_{in}$, $L_0$, the polymer

**Fig. 5 | Characterization of pneumatic response performance of twisted CLCE hollow fiber. a** Schematic diagram of the programming process for obtaining CLCE hollow fiber with twisted alignment. **b** POM images of left-handed twisted CLCE hollow fibers with $\tau_0$ of 9° mm$^{-1}$ (top) and 18° mm$^{-1}$ (down), respectively. **c** XRD patterns (left) and intensity of azimuthal scan (right) of left-handed twisted CLCE hollow fibers with $\tau_0$ of 9° mm$^{-1}$ and 18° mm$^{-1}$. **d** Images of a red CLCE hollow fiber with 18° mm$^{-1}$ left-handed twisted alignment inflated at different pressures. **e** The rotation angle of the twisted CLCE hollow fiber as a function of pressure. **f** The axial deformation ratio of the twisted CLCE hollow fiber as a function of pressure.

**g** Reflectance spectra of the CLCE hollow fiber corresponding to **d**. **h** The reflection wavelength of the twisted CLCE hollow fiber at varying pressures. **i** Cyclic performance of rotation and color change of CLCE hollow fiber with 18° mm$^{-1}$ left-handed twist under pneumatic actuation at 140 kPa. The rotation angle of the first cycle is denoted as $\phi_0$, and the rotation angle of the x-th cycle is denoted as $\phi_x$. The stability of the rotation angle is characterized by $\phi_x/\phi_0$. **j** Simulation results of the mesogen director field, deformation, and color change for the CLCE hollow fiber with 18° mm$^{-1}$ left-handed twisted alignment under different pressures. **k** Director rotation angle as a function of pressure.

anisotropy $r$, and the non-ideality parameter $\alpha$ are chosen to be the experimental values or fitted from the monodomain experiment independently (or close). Due to the inherent symmetry of our model, the theoretical solutions for left- and right-handed twists are symmetric as well. As shown in Fig. 5e, f, h, and k, the numerical predictions from our model closely match the experimental pneumatic actuations, particularly capturing the twist/reverse-twist transitions and the jumps in length. Additionally, the predicted thickness changes enable computation of the resulting color shifts, which show good agreement with experimental observations. A further examination based on the theoretical model shows that the director reorientation caused by the hoop stress fundamentally governs the twist/reverse twist behavior and the length jump. From the above results, it can be concluded that upon inflation, the twisted CLCE hollow fiber first underwent torsional rotation and axial contraction. At $\Delta p_c$, it untwisted and reversed its rotation, exhibiting both axial contraction and radial expansion. Beyond $\Delta p_c$, the fiber exhibited axial elongation, accompanied by a blueshift in color.

In summary, by utilizing dynamic covalent bonds, CLCE hollow fibers with designed twisted structures can be achieved. Under pneumatic actuation, due to the twisted alignment of mesogens, the CLCE hollow fiber exhibited not only length contraction but also torsional deformation. Notably, the entire process involved two-directional motion, including twisting and untwisting rotations, with higher twist densities resulting in greater overall rotation angles. Additionally, the pneumatic actuation process was accompanied by a blueshift in structural color, providing responses in both shape and color. This twisted CLCE hollow fiber enriches the deformation modes of CLCE fibers as actuation units and expands the application scenarios of CLCEs as stimuli-responsive materials.

## Discussion

In this study, polydomain CLCE hollow fibers with periodic helical nanostructures were fabricated using an anisotropic deswelling-assisted template method. By introducing dynamic covalent boronic ester bonds, the CLCE hollow fibers could be programmed with designed alignments in three dimensions—longitudinal, circumferential, and twisted orientations. Thanks to the intrinsic mechanical anisotropy, the CLCE hollow fibers could respond to pneumatic actuation, not only generating specific motions such as expansion, contraction, elongation, and twisting, but also exhibiting dynamic color changes across the entire visible spectrum. More surprisingly, most of these motions are non-monotonic, which are observed in some LCE systems[53,54] but rarely seen in conventional soft material systems. The non-monotonicity arises from a sub-critical instability associated with the director rotation in CLCEs, in contrast to the super-critical instability for inflating a normal balloon. Through a unified theoretical model and the corresponding numerical simulations, the mechanism of the pneumatic actuation response of the CLCE hollow fibers was deeply elucidated, visually demonstrating the relationship between the rotation of the LC director and the actuation behavior. It can be concluded from the theory and the experiment that the coupling of director programming, director reorientation and elastic (ballooning) instability governs the multimodal actuations together. This is of significant importance for understanding the properties arising from the intrinsic characteristics of LC materials. Overall, this

work advanced the alignment techniques for CLCE fibers. By combining mechanical/pneumatic force induction with dynamic covalent B−O bonds, CLCE fibers could be easily programmed with multiple designed alignments. Notably, the successful achievement of circumferential alignment in CLCE fibers, which had never been achieved before, marks a breakthrough in the field of CLCEs and even LCEs more broadly. In addition, this work revolutionized the actuation mechanism of CLCEs. We introduced the intrinsic properties—mechanical anisotropy and soft elasticity—into the mechanochromic process of CLCEs. Adaptive deformation and color change were achieved through force-induced reorientation of the LC mesogens, rather than relying on the traditional thermally induced LC-to-isotropic phase transition. Furthermore, this study enriched functionality. Leveraging programmed alignments and pneumatic actuation, multimodal actuation with synchronized color response was achieved. This enables CLCE fibers to transition from single-function optical devices to multifunctional actuators, providing an ideal platform for the development of smart textiles and soft robotics[55–62]. Moreover, on the theory side, this study deepens our understanding of the interaction between the mechanical response and the CLCE material property. Therefore, the study of pneumatic CLCE hollow fibers promotes the development of CLCEs as an advanced stimuli-responsive material in fields such as flexible devices, smart actuation, and adaptive systems[63–68].

## Methods
### Materials
1,4-bis-[4-(3-acryloyloxypropyloxy)benzoyloxy]−2-methylbenzene (RM257) (97%) was purchased from Chemfish Co., Ltd. The chiral dopant LC756 (95%) was purchased from Nanjing Xinyao Technology Co., Ltd. 3,6-dioxa-1,8-octanedithiol (DODT) (98%), pentaerythritol tetra(3-mercaptopropionate) (PETMP) (95%), 1-hydroxycyclohexyl phenyl ketone (I-184) (99%), benzene-1,4-diboronic acid (98%), 1-thioglycerol (99%), and ethanol (99.5%) were purchased from Adamas. Dipropylamine (DPA) (98%) was obtained from TCI. Toluene was purchased from TGREAG. Magnesium sulfate and heptane were obtained from Greagent. Tetrahydrofuran was purchased from Energy Chemical. UV-curable adhesive (8500 Metal) was purchased from Ergo. The silicone tube and capillary glass tube are commercially purchased products. The aforementioned materials were used as received without any additional purification.

### Synthesis of 2,2'-(1,4-phenylene)-bis[4-mercaptan-1,3,2-dioxaborolane] (BDB)
BDB was synthesized following a previously reported method[38]. Benzene-1,4-diboronic acid (1.5 g, 9 mmol) and 1-thioglycerol (2.0 g, 18.5 mmol) were dissolved in a mixture of tetrahydrofuran (40 mL) and water (0.1 mL). Magnesium sulfate (4.0 g) was then added. After stirring at room temperature for 24 h, the mixture was filtered and concentrated. The resulting solid was subsequently purified by repeated cycles of filtration and washing with abundant heptane, followed by concentration to afford the target compound as a white solid.

### Fabrication of CLCE hollow fiber
In the first step, 0.6025 g of RM257 and 0.0174 g of LC756 were weighed into a glass vial, with LC756 accounting for 2.8 wt% of the total

mass of the LC monomers. Then, 450 μL of toluene was added, and the mixture was heated to 80 °C to dissolve completely. After cooling to room temperature, 0.1265 g of DODT, 0.0527 g of BDB, 0.0386 g of PETMP, and 0.0041 g of I-184 were added. Finally, 0.100 g of a DPA solution diluted with toluene at a volume ratio of 1:50 was added and thoroughly mixed to form the precursor.

In the second step, a mold was assembled by fitting silicone tube I ($\Phi_{in}$: 2.2 mm, $\Phi_{out}$: 3.2 mm) over a capillary glass tube ($\Phi_{in}$: 0.6 mm, $\Phi_{out}$: 1.3 mm). Approximately 1 cm of silicone tube II ($\Phi_{in}$: 1.4 mm, $\Phi_{out}$: 2.2 mm) was inserted at one end of the mold as a spacer to align the centers of silicone tube I and the capillary glass tube, ensuring uniform wall thickness of the resulting hollow fiber. The LC precursor was then vacuum-filled into the mold from the other end. Upon completion, another silicone tube II was inserted at this end. The mold was placed in a fume hood for 12 h to allow solvent evaporation and the proceeding of the first-stage Michael addition reaction.

In the third step, the mold was exposed to 365 nm UV light at an intensity of 1.0 mW cm$^{-2}$ for 0.5 h to complete the second-stage photocrosslinking. After this, the sample was soaked in ethanol for 4 h to facilitate its removal from the mold. Finally, the ethanol was completely evaporated to obtain the CLCE hollow fiber.

The CLCE hollow fibers fabricated using the proportions outlined in the above experimental process exhibited reflection wavelengths within the NIR range. To obtain red CLCE hollow fibers, the LC756 content was adjusted to 4.0 wt% of the total LC monomer mass. For green and blue CLCE hollow fibers, the LC756 content was 4.9 wt% and 5.3 wt%, respectively.

### The orientation process of the CLCE hollow fiber
(i) For longitudinally aligned CLCE hollow fiber: The obtained polydomain CLCE hollow fiber with a NIR wavelength was stretched to twice its original length and fixed, at which point it showed a red color. It was then placed in an oven at 65 °C for 6 h to allow for the exchange of dynamic B−O bonds, fixing the orientation state of the fiber. After cooling to room temperature and removing from the fixture, a red, longitudinally oriented CLCE hollow fiber was obtained. (ii) For circumferentially aligned CLCE hollow fiber: The longitudinally oriented CLCE hollow fiber was placed in an oven at 65 °C and connected to an air pump for inflation at increments of 10 kPa. As the air pressure increased to a certain value, the length of the CLCE hollow fiber decreased. When the length of the CLCE hollow fiber reached its minimum, the air pressure was maintained, and it was left in the oven at 65 °C for 6 h to allow for the rearrangement of the dynamic covalent network and fixation of the orientation of the LC mesogens. After cooling to room temperature, a circumferentially oriented CLCE hollow fiber was obtained. (iii) For twisted CLCE hollow fiber: The obtained polydomain CLCE hollow fiber with an NIR wavelength was stretched to twice its original length while being twisted. In this work, samples with a length of 2 cm were subjected to 180° and 360° left-handed twists, as well as 180° and 360° right-handed twists. After being placed in an oven at 65 °C for 6 h to program the orientation, twisted CLCE hollow fibers with varying twist densities and chiralities were obtained.

### Characterization of CLCE hollow fiber
[1]H NMR of BDB was collected by a nuclear magnetic resonance apparatus (Ascend TM 400 MHz, Bruker) with samples dissolved in CDCl$_3$. POM images of CLCE hollow fiber were obtained using a Nikon Eclipse Ti microscope equipped with a digital camera (Nikon DS-U3), captured between crossed polarizers in transmission mode. DSC was performed using a TA Q2000 DSC system. Samples were heated from −40 to 140 °C at a heating rate of 20 °C min$^{-1}$. XRD patterns of CLCE hollow fiber were recorded using a Bruker D8 Discover diffractometer. The stress-strain test was conducted using an Instron 3342 tension tester at a tensile rate of 5 mm min$^{-1}$. DMA tests were conducted on a TA Q800

DMA system. The samples were stretched at a force rate of 1.0 N min$^{-1}$ until failure to obtain stress-strain curves at temperatures of 25, 45, 65, and 85 °C. The near-linear portion within 10% strain was used to calculate the Young's modulus. For stress relaxation tests, samples with different BDB contents were rapidly stretched to a strain of 5% and held at 65 °C. Reflection spectra were recorded using a Wyoptics PC2000 spectrometer. The SEM images were obtained by a field emission SEM (SU-8010, Hitachi).

### Characterization of the pneumatic response performance
One end of the CLCE hollow fiber sample was inserted into a needle (19 G or 21 G, selected based on the sample size) and fixed with UV-curable adhesive. The needle was connected to an air pump. The other end was sealed with UV-curable adhesive, and a 0.6 g iron clamp was attached to the tip. The sample was pneumatically actuated by applying the pressure in increments of 10 kPa, with the maximum pressure adjusted according to the sample's condition to prevent bursting. After each pressure adjustment, the sample was allowed to stabilize, and its shape and color were recorded using a camera.

### Theory
Here we model the CLCE hollow fiber in $(z, \theta, \rho)$ cylindrical coordinates, with initial inner radius $R_{in}$, outer radius $R_{out}$, length $L_0$ and thickness $t_0$. Upon inflation, the balloon stretches $L_0 \to \lambda L_0$, dilates $R_0 \to \eta R_0$, and consequently thins $t_0 \to t_0/(\lambda\eta)$ to conserve the elastomer's volume. The deformation gradient **F** in the CLCE is thus given by

$$\mathbf{F} = \begin{bmatrix} \lambda & 0 & 0 \\ \gamma & \eta & 0 \\ 0 & 0 & 1/(\lambda\eta) \end{bmatrix}, \qquad (2)$$

in the cylindrical coordinates, where the shear $\gamma$ is canceled out in the polydomain, longitudinal and circumferential cases but survives in the helical case.

We then adopt the non-ideal neo-classical hyper-elastic energy density[3], shown in Eq. (1), to simulate the ballooning of CLCE fibers, where $\mu$ is the shear modulus, $r > 1$ is the polymer anisotropy and $\alpha$ is the small non-ideality. In the following simulations, $\mu$, $r$ and $\alpha$ are the only three material parameters employed. Here, $\mathbf{n}_0$ is the anchored director during programming. In our programmed CLCE case, $\mathbf{n}_0$ remains on the $(z, \theta)$ plane but twists through the thickness, given by $\mathbf{n}_0 = (\cos\theta(\rho), \sin\theta(\rho), 0)$. The rotation angle $\theta(\rho)$ is set to be $\theta(\rho) = \theta_0 + \delta \sin(2\pi\rho/p)$, where $\theta_0$ is determined by the second crosslink (e.g. $\theta_0 = \pi/2$ for the circumferential pattern), $\delta$ in the second term is a small number indicating the director oscillation through the thickness, and $p$ is the pitch of the twisted director in CLCEs which is given by $h/N_p$ (thickness/number of pitches). In the twist alignment case, the angle $\theta_0$ is obtained by the measured bias angle as mentioned in the main text.

Applying the homogeneous deformation gradient **F** across the thickness, the overall elastic membrane energy per unit area is given by an integral over the thickness $\rho$:

$$\widetilde{W}(\lambda, \eta, \gamma) = \int_{R_{in}}^{R_{out}} W(\mathbf{F}, \mathbf{n}_0) d\rho. \qquad (3)$$

Since the thickness is an integer multiple of the pitch, the integration cancels out the pitch dependence and depends only on the thickness. Accordingly, the total energy, including the elastic energy and the pressure-volume work, is given by

$$W_{total}(P, \lambda, \eta, \gamma) = \pi\left(R_{out}^2 - R_{in}^2\right) L_0 W(\mathbf{F}, \mathbf{n}_0) - \pi R_{in}^2 L_0 \lambda\eta^2 P. \qquad (4)$$

The inner radius $R_{in}$, outer radius $R_{out}$, and initial length $L_0$ are obtained from experiments (Table 1). We then fix the pressure $P$ and minimize the total energy to obtain $(\lambda, \eta, \gamma)$ and the corresponding

**Table 1 | The approximate initial dimensions of the CLCE hollow fiber samples**

| Samples | Inner diameter (mm) | Outer diameter (mm) | Wall thickness (mm) | Length (mm) |
|---|---|---|---|---|
| Polydomain | 1.3 | 2.2 | 0.45 | 16 |
| Longitudinal alignment | 1.0 | 1.7 | 0.35 | 18 |
| Circumferential alignment | 2.3 | 3.0 | 0.35 | 13 |
| Twist alignment | 1.0 | 1.7 | 0.35 | 15 |

deformation gradient $\mathbf{F}$. The twisting angle $\phi$ in Fig. 5 can be obtained from $\gamma$ as $\phi = \frac{\gamma L_0}{\eta R_{out}}$. The color change is directly related to the thickness change. According to the reported literature[3,69], the measured wavelength $\lambda_{RT}$ is a linear function of the thickness $t$ by $\lambda_{RT} = n_{ave} t/N_p$, where $n_{ave}$ is the average refractive index and $N_p$ is the number of pitches. Thus, the theoretical wavelength is given by $\lambda_{RT} \propto t_0/(\lambda\eta)$. With all the deformation parameters and wavelength shifts obtained, the theoretical configurations during inflation can be obtained (e.g. Figs. 2j, 3j, 4j, 5j).

In the four experiments, we perform a similar minimization with different distributions of $\theta(\rho)$ to obtain $\lambda, \eta, \gamma$ for a given pressure $P$, and then plot the length strain, diameter strain and twist angle against pressure. It should be noted that, in our simulations, the geometrical parameters $R_{out}, R_{in}, L_0$ are obtained from the experimental data (Table 1) and the material parameters (the polymer anisotropy $r$, non-ideality parameter $\alpha$ and shear modulus) are fitted from the monodomain experiment independently (or close). As shown in the figures, our methods successfully capture the key features of the experiments, indicating that they are grounded in the correct underlying physics.

The geometric data of the fibers are given in Table 1. The distribution $\theta(\rho)$ for the four experiments and the material parameters are

1. Polydomain. $\theta(\rho) = 2\pi\rho/p$, $r = 5.5$, $\alpha = 0.05$, $\mu = 350$ kPa.
2. Longitudinal alignment. $\theta(\rho) = 0.15 \sin(2\pi\rho/p)$, $r = 5.5$, $\alpha = 0.05$, $\mu = 350$ kPa.
3. Circumferential alignment. $\theta(\rho) = \pi/2 + 0.05 \sin(2\pi\rho/p)$, $r = 5.5$, $\alpha = 0.05$, $\mu = 350$ kPa.
4. Twist alignment.
   (a) 18-Right. $\theta(\rho) = \pi/12.2 + 0.2 \sin(2\pi\rho/p)$, $r = 5.5$, $\alpha = 0.05$, $\mu = 370$ kPa.
   (b) 9-Right. $\theta(\rho) = \pi/28.1 + 0.2 \sin(2\pi\rho/p)$, $r = 5.5$, $\alpha = 0.05$, $\mu = 400$ kPa.

To better account for the anisotropic modulus—which is expected to yield improved predictions in the circumferential and the monodomain case—we adopt an alternative model (Theory-2 in Fig. 4f) that captures both semi-softness and modulus anisotropy[51]. The corresponding hyper-elastic energy density is formulated as

$$W(\mathbf{F}, \mathbf{n}_0) = \min_{\mathbf{n}} \left[ \frac{1}{2}\mu_1 \mathrm{Tr}\left( \mathbf{F}^T \ell^{-1}(\mathbf{n})\mathbf{F}\ell(\mathbf{n}_0) + \alpha\mathbf{F}(\mathbf{I} - \mathbf{n}_0\mathbf{n}_0)\mathbf{F}^T\mathbf{nn} \right) + \frac{1}{2}\mu_2 \left( \mathrm{Tr}\left( \ell^{1/2}(\mathbf{n}_0)\mathbf{F}^T \ell^{-1}(\mathbf{n})\mathbf{F}\ell^{1/2}(\mathbf{n}_0)\mathbf{n}_0\mathbf{n}_0 \right) - 1 \right)^2 \right], \quad (5)$$

where the first term is the same as the previous model, while the second term incorporates the anisotropy (a smaller $\mu_2$ corresponds to a smaller modulus perpendicular to the director). Specifically, the material parameters in the model (Theory-2 in Fig. 4f) are $r = 5.5$, $\alpha = 0.05$, $\mu_1 = 350$ kPa, $\mu_2 = 122.5$ kPa, and for the circumferential case $\theta(\rho) = \pi/2 + 0.05 \sin(2\pi\rho/p)$.

## Data availability

The data supporting the findings of this study are available within the Article and its Supplementary Information. Source data are available on Figshare: https://doi.org/10.6084/m9.figshare.31417376. All data are also available from the corresponding author upon request.

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

## Acknowledgements

Z.Y. acknowledges the financial support from the National Natural Science Foundation of China (Grant No. 52573036) and the Tsinghua University Dushi Program. F.F. acknowledges the financial support from the National Natural Science Foundation of China (Grant No. 12472061). J.B. received funding from a UKRI Future Leaders Fellowship (Grant Nos. MR/S017186/1 and MR/Y033957/1).

## Author contributions

Z.Y. and J.M. conceived the project. J.M. conducted the experiments and wrote the manuscript. J.B. and F.F. conducted theoretical calculations. J.B., F.F., and Z.Y. wrote and revised the manuscript. All authors discussed the results and approved the final version of the manuscript.

## Competing interests

The authors declare no competing interests.
