## [Transparent Peer Review file · Nature Communications]

Programmable multimodal actuation in cholesteric liquid crystal elastomer hollow fibers beyond mechanochromism

Corresponding Author: Professor Zhongqiang Yang

Version 0:

Reviewer comments:

Reviewer #1

(Remarks to the Author)

This work presents new findings suggesting that programming nematic orientation in hollow CLCEs (and the resulting anisotropy) enables multimodal pneumatic actuations. While I acknowledge that this is an interesting strategy/observation, I have a few reservations that need to be clearly addressed. In my opinion, even if these question marks are resolved unambiguously, this work still lacks a level of significance or profound implication that meets the high standard of Nature Communications, from either a fundamental or an application perspective. My comments are as below:

(1) My major concern is the authors' claim that the reactive mesogens rotate from a longitudinal (or polydomain) alignment to a circumferential alignment during pneumatic inflation. In my view, this is overstated. The schematic illustration in this paper implicitly assumes the bistability of reactive mesogens, i.e., mesogens adopt either longitudinal or circumferential alignment. However, there is no basis to think that such discrete states can be energetically stable, nor that inflation alone can drive a well-defined transition between them. I would agree with a partial director rotation, but it seems too dramatic to assert that the mesogens can undergo a definite 90° rotation while being chemically crosslinked. If dynamic bond exchange plays any role (which I presume is unlikely), the reported cyclability would be difficult to reconcile.

(2) Figure 4d, WAXS data, is presented as the key data supporting the author's claim that circumferential alignment is observed after inflation. This is the part where my reservation arises, because the X-ray image is rotated by 90° relative to other WAXS patterns such as Figure 3d (I can tell this by the position of the beam stopper). This is very critical.

(3) I strongly recommend that the authors perform in-situ experiments to directly observe the alignment rotation during inflation.

(4) The authors define that the Δp_c for the CLCEs with a longitudinal alignment is 80 kPa, but I disagree. The Δp_c appears to be rather 60 kPa.

(5) I disagree with the statement on page 12 line 285-288. A drastic change in the contraction is concurrent with the wavelength. I can see a linear trend from 60 kPa to 120 kPa.

(6) I disagree with the statement on page 9 line 217-218. The agreement between experimental results and theoretical simulations is more than a little disappointing. I think the theory is quite off from reality (Figure 2e & Figure 2g). Figure 2g has obviously a linear trend.

(7) How do the helical structures behave as longitudinal or circumferential alignment occur? Detailed discussion is required, possibly based on experimental data.

(8) The degree of alignment in CLCEs does seem to have a distinct impact on anisotropic behaviors of CLCEs during pneumatic actuation. It determines the range of pressures that the CLCEs can resist, benefited from the anisotropic nematic ordering. I think that though it is narrow, the CLCEs with polydomains also seemingly have that persistent (plateau) pressure range, from 0 to 20 kPa. The correlation between the alignment and the anisotropic behaviors of CLCEs is indeed new knowledge that this work brings; however, I believe that this finding is not sufficiently novel or unprecedented to warrant the publication in Nature Communications.

(9) There are many recent works related to hollow CLCE fibers (e.g. Adv. Mater. 2025, 37, 2504461; Adv. Sci. 2025, 12, e04487). I think that relevant work should be well acknowledged and properly compared with the current work in the introduction. In particular, while the deswelling-assisted template method in this work seems the identical method used in Adv. Sci. 2025, 12, e04487, the authors did not properly mention the prior work.

(10) The authors should use clear and straightforward terminology throughout the manuscript. For example, on page 1 line 21, nematic phases are not parents of cholesteric phases (i.e., a cholesteric phase is not a subphase of nematic phase).

Reviewer #2

(Remarks to the Author)

This work presents a significant advancement in the field of soft smart materials by developing programmable cholesteric liquid crystal elastomer (CLCE) hollow fibers with multimodal pneumatic actuation and synchronous color changes. The authors successfully combine dynamic covalent chemistry, director programming, and pneumatic control to achieve complex deformation modes, which are coupled with dynamic structural color shifts. To understand these phenomena theoretically, the authors develop a unified theoretical model using non-ideal LCE energy and solve a ballooning instability problem. The theoretical results agree with the experiments well, revealing that the instability involving director rotation plays a significant role. This work sets a new benchmark for the field of adaptive color-changing and intelligent actuation systems. Overall, the paper is well-structured and written. The methodology is robust, the results are compelling, and the theoretical framework provides valuable physical insight. The topic and the quality match the requirements of Nature Communications. I highly recommend its publication.

Here are some comments for authors to consider and further improve the work.

1. During the programming process, the CLCE hollow fiber needs to undergo stretching, twisting, or pneumatic expansion to align the liquid crystal mesogens. Providing the mechanical properties of the CLCE hollow fiber before programming is crucial for completing the experimental information.
2. For the polydomain sample, which lacks mechanical anisotropy and directional orientation, why does the CLCE hollow fiber initially undergo axial contraction rather than expansion upon inflation?
3. The current work focuses on pneumatic actuation, which requires a physical connection to an air pump. Is it possible to achieve wireless or remote actuation to free the system from such constraints?
4. The authors have developed a method combining dynamic covalent bonds with mechanical/pneumatic fields to achieve various alignment programming of CLCE hollow fibers. Conceptually and practically, what are the main breakthroughs and innovations of this strategy compared to traditional alignment methods for CLCE or LCE fibers?
5. We've seen some work on the ballooning instability of LCE balloons. Please clarify what are the differences between the inflation of an LCE balloon and a CLCE balloon.
6. In Figure 3 (longitudinal alignment), the wavelength keeps almost unchanged in a relatively large range of inflation. What is the mechanism behind this phenomenon? Could this point be used to design an actuator or some other applications?

Reviewer #3

(Remarks to the Author)

This was a very entertaining paper, and I found the results interesting. Achieving the alignments in the LC balloons was noteworthy, and the study was quite thoroughly done. I have few comments:

- Figure 1g. I suggest the authors state in the legend what the arrows represent. It is a bit of a difficult story, and it would help a lot if the authors distinctly state that the arrows delineate the mesogen alignment and not, for example, the helical axes.
- This may be a naïve question, but I ask it anyway. Is the mesogens in the polymerized film are forced to rotate towards the longitudinal direction, I assume this force is felt all through the depth of the helix. As the individual mesogens collectively rotate along the longitudinal, does this not affect the reflection band? I would assume this lowers the pitch, resulting in a red shift. However, the thickness decreases, tightening the pitch. Which feature wins out under extreme extension?
- Ln 266: "while maintaining enough residual wobble alignment to preserve the structural color" Is there any estimation of the % reflection from the fibers? The only plot I see has 'relative' values. Can the authors estimate the fractional reflection?

Version 1:

Reviewer comments:

Reviewer #1

(Remarks to the Author)

Authors clarified my questions properly.

Reviewer #2

(Remarks to the Author)

The author has finely addressed all of the issues. Now the manuscript has been improved and can be accepted.

Reviewer #3

(Remarks to the Author)

I have read the author's responses to all reviewers, and I am satisfied they have made the alterations that were required of them. I still find it an interesting work and is deserving of publication.

1 February 2026

Dear Dr. Virginia Valderrey Berciano,

Manuscript ID: NCOMMS-25-87415-T

Title: “Beyond mechanochromism: Programmable multimodal actuation in cholesteric liquid crystal elastomer hollow fibers”

We have carefully considered reviewers’ comments and made revisions (highlighted in bright yellow in the revised manuscript) as detailed below.

Reviewer #1

Comments: This work presents new findings suggesting that programming nematic orientation in hollow CLCEs (and the resulting anisotropy) enables multimodal pneumatic actuations. While I acknowledge that this is an interesting strategy/observation, I have a few reservations that need to be clearly addressed. In my opinion, even if these question marks are resolved unambiguously, this work still lacks a level of significance or profound implication that meets the high standard of *Nature Communications*, from either a fundamental or an application perspective.

Reply: We truly appreciate that the reviewer acknowledges that our work is interesting, and also the care with which they have scrutinized the manuscript. We hope the referee’s technical questions are addressed clearly below. However, we must challenge the idea that our work lacks the level of significance required for *Nature Communications*. We believe our work is a very significant step forward for cholesteric liquid crystal elastomers, as we show, for the first time, how their shifting structural colors may be combined with the anisotropic soft elasticity and director rotation of nematic LCEs. This work thus unites two quite different aspects of LCE science—cholesteric color and nematic soft elasticity—making it a fundamental contribution to both fields.

As might be expected, this fundamentally new combination leads directly to several new observations and opportunities:

(1) Soft elasticity enables a new fabrication strategy for CLCE fibers with an overall director—namely mechanical programming *via* inflation/force followed by dynamic boronic ester bond exchange. This strategy allows us to make a suite of fibers with different overall orientations, including circumferential overall alignment, which has been a longstanding challenge in the CLCE fiber field.

(2) The resulting fibers show a range of unusual deformation pathways upon inflation, including twisting, anomalous contraction, and instabilities. These are understood as a consequence of the soft elasticity, and connect us with an extensive literature on such instabilities/deformations in nematic elastomer balloons. Moreover, we not only connect CLCEs with this nematic soft-elastic literature; we actually significantly exceed the nematic literature in both theory and observation—including the first observation of a sudden twisting instability under oblique alignment, and the first presentation of the corresponding mechanical theory.

(3) Finally, all these complex deformation pathways are accompanied by color changes that we are also able to predict and understand. Our system thus delivers true multifunctionality with original color, color-pathway and deformation pathway all subject to design. Furthermore, we can imagine the same level of multi-effect design being applied to any CLCE structure, not just the fibers studied here. Our work is thus the foundation for a new generation of CLCE experiments and devices. We believe these three factors clearly bring our work to the level of *Nature Communications*.

Comment 1: My comments are as below: My major concern is the authors' claim that the reactive mesogens rotate from a longitudinal (or polydomain) alignment to a circumferential alignment during pneumatic inflation. In my view, this is overstated. The schematic illustration in this paper implicitly assumes the bistability of reactive mesogens, i.e., mesogens adopt either longitudinal or circumferential alignment. However, there is no basis to think that such discrete states can be energetically stable, nor that inflation alone can drive a well-defined transition between them. I would agree with a partial director rotation, but it seems too dramatic to assert that the mesogens can undergo a definite 90° rotation while

being chemically crosslinked. If dynamic bond exchange plays any role (which I presume is unlikely), the reported cyclability would be difficult to reconcile.

Reply 1: We thank the reviewer for raising this concern. Ultimately, the existence of large director rotations, and their effect on the CLCEs' mechanics, are the key concepts in the paper—and, in our view, the fact the reviewer doubts their possibility underlines the novelty and importance of our work.

We first highlight that such large (90-degree) rotations of the director are a commonplace and accepted feature of nematic elastomers. This has been understood since the pioneering work of Warner and Finkelmann. Below we show a classic figure (**Figure R1**) showing director rotation angle (left) and stress (right) against stretch during a tensile stretching experiment. In particular, we highlight how the director does not just asymptote towards 90-degree rotation at large stretch, but rather reaches the full 90 degrees at a large but finite stretch.

Figure R1. The director rotation angle $\theta(\lambda)$ in each domain (left) and an associated stress-strain curve showing the soft-elasticity plateau in a sample of LCE with flexible tri-functional cross-links (right) (*J. Phys.: Condens. Matter* **1999**, *11*, R239).

From a theoretical perspective this behavior is very well predicted by the now well accepted non-ideal neo-classical energy model for nematic elastomers, which was originally derived (from microscopic statistical mechanics) by Warner, and which was used to create the theory lines on the above plots. The energy density is given by (also shown in the main text)

$$W(\mathbf{F}, \mathbf{n}_0) = \min_n \left[\frac{1}{2} \mu \text{Tr}(\mathbf{F}^T \boldsymbol{\ell}^{-1}(\mathbf{n}) \mathbf{F} \boldsymbol{\ell}(\mathbf{n}_0)) + \alpha \mathbf{F}(\mathbf{I} - \mathbf{n}_0 \mathbf{n}_0) \mathbf{F}^T \mathbf{n} \mathbf{n} \right],$$

$$\text{where } \boldsymbol{\ell}(\mathbf{n}) \equiv r^{-1/3}(\mathbf{I} + (r-1)\mathbf{n}\mathbf{n})$$

Moving beyond tensile testing, there is also a more modern literature on the inflation of cylindrical nematic elastomer balloons. Inflation is not the same as tensile testing, as it involves a biaxial stress state rather than a uniaxial one. However, the fundamentals of force balance require that the circumferential stress is twice the longitudinal stress, and the above energy still predicts director rotation from longitudinal to circumferential. This conclusion has been observed experimentally (*J. Mech. Phys. Solids* **2020**, *142*, 104013), explored theoretically (*J. Appl. Phys.* **2021**, *129*, 114701; *EPL*, **2020**, *132*, 36001) and observed in finite elements (A.E.H. Chehade, B. Shen, C.M. Yakaki, T.D. Nguyen, S. Govindjee, “*Finite element modeling of viscoelastic liquid crystal elastomers*,” UCB/SEMM-2023/01, University of California Berkeley (2023). <https://escholarship.org/uc/item/64d1w46t>) (**Figure R2**). Additionally, even traditional isotropic balloons often show the classic ballooning instability under pressure-controlled inflation, leading to a jump in volume/strain at a critical pressure. In the context of longitudinal nematic elastomer balloons, it has been observed (*J. Mech. Phys. Solids* **2020**, *142*, 104013) and explained (*EPL*, **2020**, *132*, 36001) that a similar instability can result in a sudden rotation of the director from longitudinal to circumferential, unlike the continuous rotation seen in tensile testing.

Figure R2. Snapshots of the (unscaled deformation) of the LCE balloon with director field and contours of axial motion at pressures of 0, 10, 20, 30, 40, 50, 60, and 70 kPa, top to bottom (A.E.H. Chehade, B. Shen, C.M. Yakaki, T.D. Nguyen,

S. Govindjee, “*Finite element modeling of viscoelastic liquid crystal elastomers,*” UCB/SEMM-2023/01, University of California Berkeley (2023). <https://escholarship.org/uc/item/64d1w46t>).

We hope these points clarify that the mesogens (directors) really can and do rotate by 90 degrees in nematic elastomers, even after crosslinking. Is the same expected in cholesteric LCEs? Cholesteric and nematic liquids differ in their free-energy by the addition of a source term q_0 in their Frank-elastic twisting energy, $(K_2(\mathbf{n} \cdot \text{curl}(\mathbf{n})))^2 > (K_2(\mathbf{n} \cdot \text{curl}(\mathbf{n}) + q_0))^2$, which makes the helical structure the energy minimum. However, after crosslinking, such Frank terms are always negligible compared to polymer elasticity, so the appropriate free energy is exactly the same as that shown above, only with the director \mathbf{n}_0 following the helical profile of the cholesteric that was present at crosslinking. This view is also well-established (e.g., Eqns. 1, 2, and 3 of *PRE* **2002**, *65*, 056614 and Ch 9 of the classic book “*Liquid Crystal Elastomers*, Clarendon Press, Oxford, **2003**” by M. Warner, E. M. Terentjev), and the resulting theory of CLCEs is the standard way of explaining their blue-shits under strain (pitch follows deformation exactly because rubber elasticity dominates Frank energy). The theory also substantially predicts that the original helical director will indeed rotate towards an imposed uniaxial strain, as seen in the graph below (**Figure R3**), which shows director angle vs z evolving from a linear helix at no stretch ($\lambda = 1$) to a much more uniform state with $\theta \approx 0$ by $\lambda = 1.25$.

Figure R3. Dependence of the orientation of the director on distance along the pitch axis for helices subjected to several different x strains of magnitude λ (*PRE*

2002, 65, 056614).

We hope the above observations leave no doubt that very large director rotation is possible in nematic and cholesteric LCEs under strain, and that in cylindrical balloons, we expect rotation towards the circumferential direction. Of course, reorientation is not perfect—as seen in the $\lambda = 1.25$ line above, there is a residual periodic wobble around the $\theta = 0$ uniform state (in the figure, about ± 22 degrees, though it falls further with increasing stretch). The key innovation in our manuscript is the use of the dynamic bonds to make such strained cholesteric states (with overall alignment and a wobble) the energy minima of the network—the character of the states themselves follows the conventional wisdom in the field for strained cholesterics.

In addition to these background observations, we report numerous experimental findings that support the existence of strong reorientation (and overall alignment) in our samples, including the rotation towards circumferential during pneumatic inflation. (i) Figure 3c shows the POM image of a longitudinally aligned CLCE hollow fiber. The sample appeared completely dark when its long axis was parallel to one of the polarizers, and became bright when rotated 45° relative to the polarizer. This is the standard optical signature of strong nematic-like alignment. Figure (4c) displays similar optical data for a circumferentially aligned sample. (ii) The nature of POM means that such microscopy cannot actually distinguish, longitudinal and circumferential alignment, but this remaining ambiguity is removed by the XRD data (Figures 3d(ii) and 4d), which unambiguously shows longitudinal and circumferential alignment respectively, and quantifies that both have strong alignment (orientation degree of 0.63 for the longitudinally aligned sample and 0.60 for circumferential). We note that partial rotation towards, say, circumferential, would not result in the POM dark directions being at $0/90$, nor in the XRD arc being centered at $0/90$.

Finally, we note that the deformation pathways of the fibers under inflation provides further very compelling evidence for full director rotation. In longitudinal balloons, we first observe anomalous shortening during inflation, which is associated with rotation towards circumferential, followed by a more conventional lengthening after rotation is complete. This very unusual behavior is predicted by our theory (and prior theories/experiments/numerics in nematic

balloons), where it is directly caused by rotation to circumferential (which drives shortening) followed by the completion of rotation, after which we observe conventional balloon lengthening.

We have revised the relevant statements in the revised main text (page 9 and 17).

Comment 2: Figure 4d, WAXS data, is presented as the key data supporting the author's claim that circumferential alignment is observed after inflation. This is the part where my reservation arises, because the X-ray image is rotated by 90° relative to other WAXS patterns such as Figure 3d (I can tell this by the position of the beam stopper). This is very critical.

Reply 2: Thanks for the reviewer's professional comments. During the testing process, to ensure the accuracy of the experimental results, the longitudinally aligned sample and the circumferentially aligned sample were oriented at a 90° difference relative to the instrument. The former was placed vertically, while the latter was placed horizontally, ensuring that the alignment direction for both samples was vertical relative to the instrument. This kept the position of the X-ray image consistent relative to the beam stopper.

The reason we did not place both samples at the same orientation during testing—which would have resulted in a 90° difference in their alignment directions and X-ray images—was exactly the presence of the beam stopper. The beam stopper causes the intensity in the azimuthal scan to be very low at its position. If the circumferentially aligned sample had also been placed vertically, the position of one of its diffraction arcs would have coincided with the beam stopper. This would have caused an interruption in the curve precisely at the peak positions (90° or 270°), adversely affecting the accuracy of the orientation degree calculation. By adjusting the orientation of the two samples during testing, we effectively avoided this issue. This ensured that the interruption caused by the beam stopper appeared at the trough of the curve (around 180°), thereby minimizing its impact on the calculation. This can be observed in the azimuthal scan intensity curve. We appreciate that this subtlety in how the data was collected could lead to a question about whether it is accurately reported, but we have double and triple checked this, and are certain that it was. In addition to our experimental records, we note that the subsequent inflation behavior of the balloons, and their thermal actuation both accord with our reported XRD data, and would be very hard to explain if the data

was in fact flipped by 90 degrees.

To avoid further confusion, we have explained the XRD process clearly in the caption of Figure 3d in the revised main text (page 20) and rotated the XRD pattern in Figure 4d so that the position of the beam stopper matches that in Figure 3d to avoid any misunderstanding.

Comment 3: I strongly recommend that the authors perform in-situ experiments to directly observe the alignment rotation during inflation.

Reply 3: Thanks for the reviewer's professional comments. This is an excellent suggestion to make our experimental results more convincing. In this work, XRD was the most direct method to in situ observe the alignment rotation during inflation. For a longitudinally aligned CLCE hollow fiber, as the pressure increased, the direction of the pair of diffraction arcs in the X-ray image was expected to rotate by 90° at the critical pressure, thereby demonstrating the formation of circumferential alignment. However, in practice, the XRD equipment required high voltage and had to be fully enclosed, making it impossible to place the sample—which was connected to tubing and an air pump—inside the chamber to perform in-situ experiments. Additionally, POM images can reflect the alignment of the CLCE hollow fiber through birefringence. When the longitudinally aligned CLCE hollow fiber was oriented at 45° to the crossed polarizers, the sample appeared bright, as shown in Figure 3c. When the LC mesogens were reoriented to the circumferential direction *via* inflation, the sample also appeared bright at the 45° position relative to the crossed polarizers, as shown in Figure 4c. Based on this, we could predict that during the rotation of the mesogens from longitudinal to circumferential direction, the POM image would transition from bright to dark and back to bright. However, this process was accompanied by changes in the sample shape, which caused the microscope to lose focus. Moreover, as this process was continuous, it was not feasible to adjust the focus in real time during pneumatic actuation, making this in situ observation impractical.

Fortunately, we were able to demonstrate the rotation of LC mesogens by in situ observing the changes in the circular polarization selectivity of the CLCE hollow fiber during inflation. This is based on the dependence of the circular polarization selectivity of CLCEs on the orientation of their mesogens. For example, the

literature (*Adv. Funct. Mater.* **2023**, *33*, 2304506) shows, in **Figure R4**, that when a polydomain CLCE film was subjected to asymmetric stretching (uniaxial stretching), its LC mesogens rotated and tended to align along the stress direction, causing distortion of the helical structure and the disappearance of circular polarization selectivity. In contrast, after symmetric stretching (biaxial stretching), the mesogens maintained their original helical arrangement and polydomain state, thus retaining circular polarization selectivity. In our experiments, as shown in **Figure R5**, the initial polydomain CLCE hollow fiber exhibited circular polarization selectivity: reflection colors were observed under no polarizer (NP) and right-handed circular polarizer (RCP) conditions, while no color was seen under left-handed circular polarizer (LCP). During the inflation process, as the pressure increased, the color under NP and RCP showed no significant change. However, under LCP, the color of the CLCE hollow fiber gradually became visible. Corresponding reflection spectra (**Figure R6**) further confirmed this result. This in situ observation indicated that the CLCE hollow fiber progressively lost its circular polarization selectivity as the pressure increased, suggesting the mesogens developed directional alignment. This is consistent with our statement in the text: “Although the polydomain CLCE hollow fiber exhibited overall mechanical isotropy, the hollow fiber structure experienced doubled circumferential (hoop) stress compared with longitudinal stress under the same pressure, leading to director reorientation toward the circumferential direction...” We thank the reviewer again for this suggestion, which has enriched our discussion on the alignment rotation during inflation.

Figure R4. Helical structure changes and photographs through RCP and LCP of CLCEs under a,b) asymmetric uniaxial and c,d) symmetric biaxial stretching in the literature (*Adv. Funct. Mater.* **2023**, *33*, 2304506).

Figure R5. Images of a red polydomain CLCE hollow fiber inflated at different pressures observed through NP, RCP, and LCP. The faint color at 120 kPa was likely due to either the thinning wall thickness making the sample more transparent, or the reflected wavelength approaching the ultraviolet region.

Figure R6. Reflection spectra of the red polydomain CLCE hollow fiber inflated at different pressures with NP, RCP and LCP, respectively.

We have added above results and discussion into the revised main text (page 7) and Supporting Information as Figure S3 and S4.

Comment 4: The authors define that the Δp_c for the CLCEs with a longitudinal alignment is 80 kPa, but I disagree. The Δp_c appears to be rather 60 kPa.

Reply 4: Thanks for the reviewer's careful reading and professional comments. In this work, the critical pressure Δp at which the CLCE hollow fiber began to undergo drastic deformation was defined as Δp_c . Our experimental observations show that no significant deformation was observed at 60 kPa, while a clear and sudden instability occurred by 80 kPa. Based on these results, we can confirm that the instability occurs between 60 kPa and 80 kPa. We have revised the main text to make the description more accurate.

We have added a more detailed discussion in the revised main text (page 12).

Comment 5: I disagree with the statement on page 12 line 285-288. A drastic change in the contraction is concurrent with the wavelength. I can see a linear trend from 60 kPa to 120 kPa.

Reply 5: Thanks for the reviewer's professional comments. This was very valuable for allowing us to explain the results more clearly. In our view, the blue-shift graph (Figure 3g) clearly shows two regimes. The first four points (0, 20, 40, and 60 kPa) show a first quasi-linear trend with a low gradient—i.e., a small amount of blue shift. The last three points (80, 100, and 120kPa) have a much steeper gradients, indicating much bluer shift. The switch between these two regimes is driven by the balloon instability at Δp_c , somewhere between 60 and 80 kPa (as above, we agree we should not have described this trend as starting at 80, but nor does it extend back to 60). The feature that we wish to draw attention to is that it is the balloon instability itself, at Δp_c , coincides with the dramatic shortening of the balloon—which suddenly almost halves in length—but this dramatic shortening is accompanied by only a mild blue-shift (about 8% wavelength shift between 60 and 80kPa, a much smaller factor than the 50% longitudinal strain), with much greater blue shift occurring afterwards, between 80 and 120kPa. This is visually very clear in Figure 3e, where the dramatic contraction is seen first, then the color starts to change. The explanation also accords well with the theory, and is the signature of the jump being driven by soft-elastic rotation—an essentially in-plane effect—rather than by thinning; this quasi-2D elasticity is a common feature of nematic semi-soft elasticity, but quite different to normal rubber elasticity. We have added above results and discussion to the revised main text (page 12 and 13).

Comment 6: I disagree with the statement on page 9 line 217-218. The agreement between experimental results and theoretical simulations is more than a little disappointing. I think the theory is quite off from reality (Figure 2e & Figure 2g). Figure 2g has obviously a linear trend.

Reply 6: We thank the reviewer for raising this question. We have removed the word “little,” and the manuscript now simply states the quantitative agreement “is somewhat disappointing”. Although we agree that the fit is indeed disappointing for this case, we would also like to emphasize that the theory is still getting a lot of things

right—we are correctly predicting non-monotonic and small length change, monotonic and large diameter change, and a large monotonic blue shift. Furthermore, the magnitudes of the blue shift and the diameter change are very close to reality, but the precise form of the curves and, most of all, the magnitude of the length change, are not correct. We also emphasize that the deformation path of this balloon is actually comparatively uninteresting, as it lacks the dramatic jumps and twists provided by longitudinal and helical alignment—and the theory works much better in these more interesting and more unexpected cases. Given how few fitting parameters are in the theory, getting such a good fit across so many different experiments is, overall, not disappointing at all, even though in this particular case we would certainly have preferred to do better. Of course, it is natural to ask why we are not getting a better fit for the polydomain case. The answer is two-fold. Firstly, it is also commonly observed that the nematic elastic energy we use is much more accurate for semi-soft (director rotating) modes of deformation than for “hard” modes of deformation, where effects such as changes in magnitude of alignment can also be important. The polydomain contains the full helical director, sampling all planar directors. This means that there are no fully soft elastic modes available—as the balloon inflates, some directors that started longitudinal can rotate to circumferential—which would alone be a soft deformation—but they are mechanically connected to further regions that started circumferential, and cannot rotate any further. This explains the comparatively stiff response of the balloon, and its lack of interesting soft-elastic instabilities, but also explains the poorer performance of the model in this case—it is optimized for the cases where soft elasticity is more important, which are in fact also the more interesting cases. Additionally, in this case, there is also substantial complexity arising from the fact that the in-plane structure is an in-plane polydomain—all the helices have the same pitch axis, but moving in the plane, one has a polydomain structure. Even in the simpler nematic case, the effective bulk constitutive law of such polydomain structures is very hard to figure out, as it depends on the precise configuration of the domains, and the extent to which they do or do not cooperate during deformation. Indeed, experimentally polydomain nematics can vary from very stiff to almost perfectly soft-elastic depending on their crosslinking history (*Macromolecules* **2009**, *42*, 4084), and there is actually no widely accepted constitutive law for modeling their

mechanical response—with the best available treatment being upper and lower bounds (*PRL* **2009**, *103*, 037802). In this context, the partial success of our theory for the polydomain case might actually be regarded as surprisingly good!

We have revised the corresponding description in the revised main text (page 9).

Comment 7: How do the helical structures behave as longitudinal or circumferential alignment occur? Detailed discussion is required, possibly based on experimental data.

Reply 7: We thank the reviewer for raising this question. As discussed in Comment 1, the programmed longitudinal or circumferential alignment is a residual periodic wobble around the longitudinal or circumferential uniform state, across the thickness direction (**Figure R4**). The experimental realization is the following. Firstly, as previously reported and shown in **Figure R5**, a polydomain CLCE exhibits circular polarization selectivity. When subjected to asymmetric forces, the LC mesogens of the CLCE tend to align in the direction of the applied force, which unwinds the helical structure and results in the loss of chiral photonic selectivity. The helical structure becomes distorted in plane but remains periodic through the thickness, with a shorter pitch due to helix compression accompanying the stretching as a result of volume conservation. On the other hand, symmetric forces maintain the inherent helical structure of the CLCE, causing only pitch contraction, and the CLCE continues to reflect circularly polarized light. Therefore, we can investigate how the helical structures behave by observing changes in the circular polarization selectivity of the CLCE hollow fiber after alignment.

(1) Longitudinal alignment was achieved by stretching a polydomain CLCE hollow fiber followed by thermal programming. The experimental results in **Figure R7** (the photographs at 0 kPa) show that the longitudinally aligned CLCE hollow fiber exhibited distinct reflection colors under NP, RCP, and LCP, indicating a loss of circular polarization selectivity. This suggests the helical structure became distorted during stretching but retained its periodic nature. Reflection spectra in **Figure R8** further confirmed this.

(2) Circumferential alignment was formed by inducing mesogen reorientation toward the circumferential direction during pneumatic actuation. For example, **Figure R5** shows the inflation process of a polydomain CLCE hollow fiber. In its initial state, reflection color was observed under NP and RCP, but not under LCP,

indicating circular polarization selectivity. During inflation, as pressure increased, the color under NP and RCP showed no significant change, but color gradually appeared under LCP. This indicated a gradual loss of circular polarization selectivity and the formation of a distorted helical structure; the blue shift in color also indicated a reduction in pitch. Corresponding reflection spectra are shown in **Figure R6**. **Figure R7** shows the inflation process of a longitudinally aligned CLCE hollow fiber. At the critical pressure—which signifies the formation of circumferential alignment—the CLCE hollow fiber exhibited reflection colors under both LCP and RCP, indicating the helical structure was in a distorted state. According to the reflection spectra (**Figure R8**), the reflection wavelength blue-shifted, confirming a pitch reduction.

In summary, by observing the circular polarization selectivity of CLCE hollow fibers with longitudinal or circumferential alignment, we concluded that the helical structure in both cases was distorted but remained periodic.

Figure R7. Images of a red CLCE hollow fiber with longitudinal alignment inflated at different pressures observed through NP, RCP, and LCP, respectively.

Figure R8. Reflection spectra of the red CLCE hollow fiber with longitudinal alignment inflated at different pressures with NP, RCP, and LCP, respectively.

After the programming of alignment, the director rotation is governed by the non-ideal neo-classical energy model (see Reply 1 and the main text). We have added above results and discussion into the revised main text (page 13) and Supporting Information as Figure S9 and S10.

Comment 8: The degree of alignment in CLCEs does seem to have a distinct impact on

anisotropic behaviors of CLCEs during pneumatic actuation. It determines the range of pressures that the CLCEs can resist, benefited from the anisotropic nematic ordering. I think that though it is narrow, the CLCEs with polydomains also seemingly have that persistent (plateau) pressure range, from 0 to 20 kPa. The correlation between the alignment and the anisotropic behaviors of CLCEs is indeed new knowledge that this work brings; however, I believe that this finding is not sufficiently novel or unprecedented to warrant the publication in *Nature Communications*.

Reply 8: Thanks for the reviewer's professional comments. In our view, our key finding is that imparting overall alignment to CLCEs enriches their mechanics with soft-elasticity (and rotation of this overall alignment) and other forms of anisotropy. As the reviewer comments, even the polydomain CLCE (that lacks overall alignment) do also show some mechanical richness, including a threshold for director longitudinal strain (though not hoop strain or blue shift) at about 20 kPa, and a non-monotonic longitudinal strain as inflation proceeds. These effects doubtless also relate to director rotation in the helices of the polydomain, but they are strongly suppressed by the fact that there is no overall alignment, so regions with favorable alignment and director rotation must always compromise with others that do not. In contrast, it is the longitudinally and helically aligned fibers that are able to access pure semi-soft modes, in which the overall director rotates towards circumferential. These global semi-soft modes enable much higher strains at lower stresses (longitudinal strains of ~50% rather than ~15%), result in dramatic subcritical jumps in balloon configuration, and, in the helical case, also result in dramatic twisting of the balloon. All these effects (large longitudinal contractions, jumps, twists) are absent in the polydomain case, and demonstrate the added richness achieved by having an overall alignment. We hope we have convinced the reviewer that this strategy—the mechanics of CLCEs with overall alignment—is a rich topic, and that, as the foundational study in the area, our paper is indeed worthy of publication in *Nature Communications*.
We have added above discussion into the revised main text (page 8).

Comment 9: There are many recent works related to hollow CLCE fibers (e.g. *Adv. Mater.* 2025, 37, 2504461; *Adv. Sci.* 2025, 12, e04487). I think that relevant work should

be well acknowledged and properly compared with the current work in the introduction. In particular, while the deswelling-assisted template method in this work seems the identical method used in *Adv. Sci.* **2025**, *12*, e04487, the authors did not properly mention the prior work.

Reply 9: Thanks for the reviewer's professional comments. In our initial submission, we had already acknowledged the articles mentioned above, citing them in the Introduction section as Reference 19 (*Adv. Sci.* **2025**, *12*, e04487) and Reference 35 (*Adv. Mater.* **2025**, *37*, 2504461). A more detailed comparison is provided below:

(1) Regarding Reference 19 (*Adv. Sci.* **2025**, *12*, e04487), in terms of fabrication, although both studies utilized the deswelling-assisted template method to prepare CLCE hollow fibers, this method could only yield polydomain structures, which is not the key claim of our article. In our work, we developed a novel method to fabricate programmed CLCE hollow fibers with longitudinal, circumferential, and twisted alignments by integrating dynamic covalent bonds with mechanical force/pneumatic pressure-induced LC mesogen orientation. This alignment technique represents a breakthrough that enables previously unattainable alignments, laying the foundation for achieving multimodal pneumatic actuation. This could not be accomplished using the deswelling-assisted template method alone.

(2) Reference 35 (*Adv. Mater.* **2025**, *37*, 2504461) studied the mechanochromic behavior of CLCE tubes under different stimuli, including stretching, inflation, and bent tube inflation. The study focused solely on color changes resulting from thickness variation and did not involve actuation deformation. In contrast, our work achieved multimodal actuation with synchronous color adaptation of CLCE hollow fibers under a single pneumatic actuation mode by utilizing diverse programmed alignments, based on mechanical anisotropy and soft elasticity. This enables CLCEs to not only work as single-function photonic devices but also possess mechanical output capabilities.

(3) In addition to the two articles mentioned by the reviewer, we also cited another paper on hollow-structure CLCEs as Reference 36 (*Adv. Funct. Mater.* **2024**, *35*, 2413965). This work also only investigated the color changes of CLCEs under different pressures.

Overall, compared with recent work on hollow CLCE fibers, whether it is the

preparation of hollow CLCE fibers or the investigation of color-changing behavior under inflation or mechanical stretching, our work was capable of achieving these. However, the programming of complex alignments in CLCE hollow fibers and the multimodal responses under pneumatic actuation were not achievable in previous work. Moreover, those three articles were limited to color changes induced by stimulus-driven thickness variation, whereas we introduced mechanical anisotropy and soft elasticity to explore new behaviors arising from force-induced LC mesogen rotation, bringing innovation at the fundamental mechanism level. Notably, two of those references were published in 2025, which fully highlights the novelty of our work and its position at the forefront of the field. We have added appropriate and relevant descriptions in the introduction of the revised manuscript (page 2).

Comment 10: The authors should use clear and straightforward terminology throughout the manuscript. For example, on page 1 line 21, nematic phases are not parents of cholesteric phases (i.e., a cholesteric phase is not a subphase of nematic phase).

Reply 10: We thank the reviewer for reminding us to revise the terminology, which makes the presentation in the manuscript more accurate. The sentence now reads: *“However, the helical structure of CLCEs averages out the exceptional anisotropy and soft elasticity of the nematic phase, leaving little scope for also using the director orientation to program their thermal or mechanical actuation.”* We have changed this terminology in the revised manuscript (page 1).

Reviewer #2

Comments: This work presents a significant advancement in the field of soft smart materials by developing programmable cholesteric liquid crystal elastomer (CLCE) hollow fibers with multimodal pneumatic actuation and synchronous color changes. The authors successfully combine dynamic covalent chemistry, director programming, and pneumatic control to achieve complex deformation modes, which are coupled with dynamic structural color shifts. To understand these phenomena theoretically, the authors develop a unified theoretical model using non-ideal LCE energy and solve a ballooning instability problem. The theoretical results agree with the experiments well, revealing that the instability involving director rotation plays a significant role. This work sets a new benchmark for the field of adaptive color-

changing and intelligent actuation systems. Overall, the paper is well-structured and written. The methodology is robust, the results are compelling, and the theoretical framework provides valuable physical insight. The topic and the quality match the requirements of *Nature Communications*. I highly recommend its publication.

Reply: We truly appreciate that the reviewer acknowledged our work and believed its topic and quality match the requirements of *Nature Communications*. We hope that the fundamental innovations and significant technological advances achieved in this work will go beyond conventional approaches to color-changing CLCEs, paving the way for future developments in soft actuators and adaptive systems.

Comment 1: Here are some comments for authors to consider and further improve the work.

During the programming process, the CLCE hollow fiber needs to undergo stretching, twisting, or pneumatic expansion to align the liquid crystal mesogens. Providing the mechanical properties of the CLCE hollow fiber before programming is crucial for completing the experimental information.

Reply 1: Thanks for the reviewer's professional comment. In this work, before programming, the CLCE hollow fiber was in a polydomain state. **Figure R9** shows the stress-strain curve of the polydomain CLCE sample. Its fracture strain was around 470%, with a corresponding stress of 2.0 MPa. The calculated Young's modulus was 0.86 MPa.

Figure R9. The stress-strain curve of the polydomain CLCE sample.

We have added above results and discussion into the revised Supporting Information as Figure S5.

Comment 2:For the polydomain sample, which lacks mechanical anisotropy and directional orientation, why does the CLCE hollow fiber initially undergo axial contraction rather than expansion upon inflation?

Reply 2: Thanks for the reviewer's professional comment, which helps us to further elaborate on the pneumatic actuation mechanism of the polydomain CLCE hollow fiber. Although the polydomain sample was mechanically isotropic, the internal stress distribution under inflation is anisotropic. In a cylindrical structure under the same pressure, the circumferential (hoop) stress is always twice the longitudinal stress—this is a feature of stress balance, independent of the material in question. Of course, both stresses are tensile, and in a conventional balloon, this does result in it getting longer and fatter during inflation. In a nematic elastomer with longitudinal alignment, this stress distribution rotates the director to circumferential in a perfect soft mode which actually shortens the balloon—this is the so-called “anomalous inflation” of nematic balloons. After the director is circumferential, the “anomalous” phase finishes, and the balloon starts to elongate classically. In contrast, in a circumferential nematic balloon, there is never any director rotation, so inflation proceeds essentially classically throughout, elongating the balloon. In the full helix present in a polydomain, we have both these cases (and every intermediate angle) present, but they must all deform in the same way, so the balloon must reach a compromise between the longitudinal case (shorten then lengthen) and the circumferential case (lengthen). The result is that the balloon does shorten then lengthen, but markedly less than with longitudinal alignment, and at much higher pressure, since the originally circumferential parts are not deforming in a soft mode.

We have added clear explanations into the revised main text (page 8).

Comment 3:The current work focuses on pneumatic actuation, which requires a physical connection to an air pump. Is it possible to achieve wireless or remote actuation to free the system from such constraints?

Reply 3: Thanks for the reviewer's professional question, which provides important guidance for the future development of pneumatic CLCE materials. The physical connection to an air pump does impose certain constraints on the actuation process.

To address this issue, one potential solution could be to incorporate phase transition materials (*Sci. Robot.* **2020**, *5*, eaaz4239; *Nat. Commun.* **2025**, *16*, 3920) or chemical reactants (*Adv. Mater.* **2024**, *36*, 2403954) into the CLCE cavity. By increasing the temperature to induce a phase transition or triggering chemical reactions to generate gas, the internal pressure could be increased for actuation. Such strategies have already been reported in some studies on pneumatic actuation. We believe that integrating these approaches with the CLCE system could contribute to pneumatic miniaturization and enhance applicability for untethered applications in the future.

We have cited the aforementioned literature as reference 66, 67, and 68 to provide support for our outlook.

Comment 4: The authors have developed a method combining dynamic covalent bonds with mechanical/pneumatic fields to achieve various alignment programming of CLCE hollow fibers. Conceptually and practically, what are the main breakthroughs and innovations of this strategy compared to traditional alignment methods for CLCE or LCE fibers?

Reply 4: We thank the reviewer for raising this question. First, conceptually, this work proposes a new alignment paradigm. Traditional methods rely on mechanical stretching, shear, or external fields, which are limited in the types of alignment they can produce within a fiber geometry. This strategy introduces a novel concept: using pneumatic pressure combined with dynamic covalent chemistry as a powerful and versatile tool for programming alignment, moving beyond the constraints of conventional techniques. Second, in practical terms, due to the limitations of previous alignment techniques, achieving circumferential alignment in a fiber had not been realized. By combining dynamic covalent bonds with mechanical/pneumatic fields, we successfully achieved circumferential alignment in CLCE hollow fibers. This represents an unprecedented breakthrough in the field of CLCEs, and even LCEs more broadly, as it had never been accomplished before. This strategy greatly expands the design space for CLCEs and significantly broadens the potential for creating mechanically anisotropic and soft-elasticity-

based devices.

We have added corresponding discussion to the revised manuscript (page 27 and 28).

Comment 5: We've seen some work on the ballooning instability of LCE balloons. Please clarify what are the differences between the inflation of an LCE balloon and a CLCE balloon.

Reply 5: We thank the reviewer for raising this question. Given the diversity of approaches that have been explored for nematic LCE balloons, it is a little complicated to describe exactly how it all relates to our work. The first paper (*J. Mech. Phys. Solids* **2020**, *142*, 104013) is an experimental and theoretical study of longitudinal nematic balloons. However, the paper is rooted in the ideal LCE energy ($\alpha = 0$) where soft modes cost no energy at all, meaning they predict full rotation to circumferential at zero pressure. To counter this, they then consider balloons with a weight hanging on the end, which keeps the director longitudinal until a pressure threshold, at which point it jumps to circumferential. They observe and predict this instability, with the weight. The paper (*Thin Wall Struct.* **2022**, *170*, 108621) does a similar theory analysis of a helical balloon with a weight, leading to a prediction of a sub-critical twisting instability, but with no corresponding experimental observation. In contrast, the papers (*EPL* **2020**, *132*, 36001; *J. Appl. Phys.* **2021**, *129*, 114701) provide a theoretical analysis of longitudinal balloons with non-ideality, such that finite pressure is required to trigger ballooning, even without a weight. These papers clarify that non-ideal nematic balloons are still expected to show a sub-critical rotation instability, but there are no corresponding experiments.

In our work, we use the classic nematic energy, with non-ideality, but also with a director profile that oscillates around the overall director in the thickness direction, reflecting the cholesteric character. Without this undulation, our theoretical model would be essentially the same as that reported for the longitudinal case (*EPL* **2020**, *132*, 36001; *J. Appl. Phys.* **2021**, *129*, 114701). This undulation is thus the key theoretical difference between the aligned CLCE balloons and pure nematic ones. The overall effect of the undulation is qualitatively similar to non-ideality, in that it also provides an energy penalty for rotation of the overall alignment, as the soft

modes of the different parts of the undulation do not exactly match, so some degree of compromise is required. This means that the instabilities of aligned CLCE balloons are not fundamentally different to those in nematic, but are delayed by this additional anchoring mechanism. In reality, this CLCE mechanism seems to be rather stronger than the true material non-ideality. As a result, our balloons show instabilities at small but finite pressures, so that we are able to observe both inflation before and after the instability, with no need to apply a weight to unnaturally delay the instability (as was done in, e.g., *J. Mech. Phys. Solids* **2020**, *142*, 104013).

We also note that, compared to the existing nematic balloon literature, we are providing a first treatment of circumferential and helical balloons with the non-ideal formulation (rather than weights), and the first corresponding experimental observations. We also note that the “undulation” that separates CLCE membranes from pure nematics exists on a spectrum, from the fully helical cases of polydomain CLCE (which is mechanically isotropic) to the fully aligned case (which is just a nematic). Thus, the undulation idea also allows us to capture the key mechanics of the polydomain case within the same theory, for which there is no nematic counterpart.

The corresponding content are presented on page 13 of the revised manuscript.

Comment 6: In Figure 3 (longitudinal alignment), the wavelength keeps almost unchanged in a relatively large range of inflation. What is the mechanism behind this phenomenon? Could this point be used to design an actuator or some other applications?

Reply 6: The mechanism here is that the balloon is deforming in a pure soft mode, wherein the director rotates from longitudinal to circumferential. The soft mode results in the polymer conformation ellipsoid rotating from longitudinal to circumferential, driving longitudinal contraction and circumferential extension, but with almost no change in thickness and hence no change in color. This two-dimensional character of soft modes is familiar in nematic elastomers under tensile tests, but unfamiliar in this context. It certainly highlights how suitable design of the overall orientation in CLCE balloons can give a degree of independent control over strain vs color-shift. We also note that the above cartoon is not perfect—there is some thinning,

and some blue shift, during the first part of actuation. Again, this is familiar from non-ideal nematic LCEs, where one does indeed see thinning prior to the onset of director rotation (the so-called semi-soft threshold), but not during director rotation. In the CLCE the residual undulation again acts like an additional non-ideality, enhancing this effect, and in future work, we plan like to study whether we can use the degree of undulation to tune this effect.

The distinct “plateau-then-jump” optical response offers unique opportunities for designing functional devices. (1) Threshold-type optical sensors. This non-monotonic effect can be used to design sensors that visually indicate when stress or stretch exceeds a set threshold—for example, a sensor that begins to change color precisely upon crossing the critical stress/stretch point. (2) Multi-stage soft actuators. In the low-pressure range, both the optical signal (structural color) and dimensions remain stable. Once the critical pressure is exceeded, a synchronous abrupt change in optical signal (color switching) and a rapid mechanical response (e.g., contraction) are triggered, enabling coordinated optical-mechanical dual-mode actuation. (3) Dynamic camouflage and information encryption. By programming CLCE fibers with different alignments, multi-mode optical responses can be achieved. Longitudinally aligned devices can “hide” color changes below the critical pressure, then suddenly display color above it. When combined with the gradual color shifts of polydomain devices, pressure-dependent dynamic pattern systems can be constructed for adaptive camouflage or encrypted information display.

The relevant discussions are presented on page 14 and 15 in the revised manuscript.

Reviewer #3

Comments: This was a very entertaining paper, and I found the results interesting. Achieving the alignments in the LC balloons was noteworthy, and the study was quite thoroughly done.

Reply: We are sincerely grateful for the reviewer's positive assessment of our work. We are particularly encouraged by the recognition of our results on alignment control in LC balloons. This technique expands the design space for CLCEs and even LCEs, providing a key foundation for leveraging mechanical anisotropy and soft elasticity, as well as for enabling multimodal actuation. This study sets a new benchmark for what CLCEs can achieve and opens avenues for their use in

advanced technological contexts. We hope this work will draw attention to and inspire further research in the field of LCE materials.

Comment 1: I have few comments: Figure 1g. I suggest the authors state in the legend what the arrows represent. It is a bit of a difficult story, and it would help a lot if the authors distinctly state that the arrows delineate the mesogen alignment and not, for example, the helical axes.

Reply 1: Thanks for the reviewer's professional suggestion, which helped us to provide a clearer schematic diagram of the multimodal actuation. We have added an arrow icon at the bottom of Figure 1g, with the text "alignment" on the right to indicate its meaning, as shown in the illustration below.

Figure R10. Morphology and working principle of CLCE hollow fiber.

Comment 2: This may be a naïve question, but I ask it anyway. If the mesogens in the polymerized film are forced to rotate towards the longitudinal direction, I assume this force is felt all through the depth of the helix. As the individual mesogens

collectively rotate along the longitudinal, does this not affect the reflection band? I would assume this lowers the pitch, resulting in a red shift. However, the thickness decreases, tightening the pitch. Which feature wins out under extreme extension?

Reply 2: We thank the reviewer for the comment, which helps clarify the thickness and color change upon director rotation. First of all, ideally the pure director rotation does not change the depth of the helix as well as the reflection band, since the director rotation is in-plane. Without director rotation, a uniaxial stretch would lower the thickness as well as the pitch, and thus cause a blue shift. With director rotation, ideally, the in-plane director rotation would preserve the thickness. For example, in the literature (*J. Phys. II France* **1997**, 7, 1059), it was observed both experimentally and theoretically that, ideally, the elastic energy remains zero, and the thickness of a thin film stays unchanged under uniaxial stretch within a certain range, predicted by the theory. However, in the case of CLCEs, the director field varies through the thickness. Directors located in certain regions of the thickness may already be aligned with the stretching direction, leading to a reduction in thickness under applied stretch. Thus, to achieve a quantitative understanding of the changes in thickness and color should be computed by our model. Please also see the detailed explanation in Comment 6 from reviewer 2.

The relevant discussions are presented on pages 12-13 of the revised manuscript, and the corresponding literature is cited as reference 49.

Comment 3:Ln 266: “while maintaining enough residual wobble alignment to preserve the structural color” Is there any estimation of the % reflection from the fibers? The only plot I see has 'relative' values. Can the authors estimate the fractional reflection?

Reply 3: We appreciate the reviewer's professional question. Regarding the % reflection of the fiber, we could measure it directly using an optical fiber spectrometer without estimation. However, since the distance between the sample and the light source affects the intensity of the reflection peak, and the deformation of the sample during pneumatic actuation causes this distance to change, we used arbitrary units (a.u.) for reflectivity to ensure more accurate data representation. By strictly controlling the position of different samples, we could indeed use percentage (%) to display the reflectivity values directly. **Figure R11** shows the reflection spectra

of the longitudinally aligned CLCE hollow fiber before and after programming. It is worth noting that although a dark reference was calibrated during testing, the inherent thickness (diameter) of the sample inevitably placed it closer to the light source than the dark reference, resulting in an increased baseline value in the curves. The fiber reflectivity is not the peak value of the curve but the difference between the peak value and the baseline. Through Lorentz fitting, the reflectivity of the fiber before and after programming was calculated to be 39% and 44%, respectively. This indicates that although stretching distorted the helical axis of the CLCE, its reflective performance was not weakened. Instead, it may have increased due to the simultaneous reflection of both left- and right-handed circularly polarized light, which is consistent with existing literatures (*Nat. Mater.* **2022**, *21*, 1441; *Adv. Sci.* **2023**, *10*, 2301414).

Figure R11. Reflectance spectra of the longitudinally aligned CLCE hollow fiber before and after programming.

We have added above results and discussion into the revised Supporting Information as Figure S7.

We appreciate the time and consideration of the editor and reviewers. We would be happy to provide further information if needed.

Yours sincerely,

Zhongqiang Yang